# Skillful Decadal Prediction of German Bight Storm Activity

Daniel Krieger[1,2], Sebastian Brune[3], Patrick Pieper[4], Ralf Weisse[1], and Johanna Baehr[3]

[1]Institute of Coastal Systems – Analysis and Modeling, Helmholtz-Zentrum Hereon, Geesthacht, Germany
[2]International Max Planck Research School on Earth System Modelling, Hamburg, Germany
[3]Institute of Oceanography, Universität Hamburg, Hamburg, Germany
[4]Institute of Meteorology, Freie Universität Berlin, Berlin, Germany

**Correspondence:** Daniel Krieger (daniel.krieger@hereon.de)

**Abstract.**

We evaluate the prediction skill of the Max-Planck-Institute Earth System Model (MPI-ESM) decadal hindcast system for German Bight storm activity (GBSA) on a multiannual to decadal scale. We define GBSA every year via the most extreme three-hourly geostrophic wind speeds, which are derived from mean sea-level pressure (MSLP) data. Our 64-member ensemble of annually initialized hindcast simulations spans the time period 1960-2018. For this period, we compare deterministically and probabilistically predicted winter MSLP anomalies and annual GBSA with a lead time of up to ten years against observations. The model produces poor deterministic predictions of GBSA and winter MSLP anomalies for individual years, but fair predictions for longer averaging periods. A similar but smaller skill difference between short and long averaging periods also emerges for probabilistic predictions of high storm activity. At long averaging periods (longer than 5 years), the model is more skillful than persistence- and climatology-based predictions. For short aggregation periods (4 years and less), probabilistic predictions are more skillful than persistence but insignificantly differ from climatological predictions. We therefore conclude that, for the German Bight, probabilistic decadal predictions (based on a large ensemble) of high storm activity are skillful for averaging periods longer than 5 years. Notably, a differentiation between low, moderate, and high storm activity is necessary to expose this skill.

## 1 Introduction

In low-lying coastal areas that are affected by mid-latitude storms, coastal protection and management may greatly benefit from predictions of storm activity on a decadal timescale. Decadal predictions bridge the gap between seasonal predictions and climate projections and may for example aid the planning of construction and maintenance projects along the coast. The German Bight in the southern North Sea represents an example of such an area, where the coastlines are heavily and frequently affected by mid-latitude storms.

Climate projections suggest that many components of the Earth system undergo changes that can be attributed to the anthropogenic global warming (IPCC, 2021). For certain types of extreme events, like heavy precipitation or heat extremes, a link between the frequency of occurrence and the change in Earth's temperature has already been established (e.g. Lehmann et al.,

2015; Suarez-Gutierrez et al., 2020; Seneviratne et al., 2021). For storm activity, studies for the past century showed a lack of significant long-term trends over the Northeast Atlantic in general and the German Bight in particular. Instead, storm activity in this region is subject to a pronounced multidecadal variability (Schmidt and von Storch, 1993; Alexandersson et al., 1998; Bärring and von Storch, 2004; Matulla et al., 2008; Feser et al., 2015; Wang et al., 2016; Krueger et al., 2019; Varino et al., 2019; Krieger et al., 2020). This dominant internal variability suggests a great potential for improved predictability in moving

from uninitialized emission-based climate projections towards initialized climate predictions. In this study, we demonstrate that initialized climate predictions are useful to predict German Bight storm activity (GBSA) on a multiannual to decadal timescale.

There have been considerable advancements in the field of decadal predictions of climate extremes in recent years. For example, the research project MiKlip (*MittelfristigeKlimaprognosen*, Marotzke et al., 2016) focused on the development of

a global decadal prediction system based on the Max-Planck-Institute Earth System Model (MPI-ESM) under CMIP5 forcing. Using experiments from the MiKlip project, Kruschke et al. (2014) and Kruschke et al. (2016) found significant positive prediction skill for cyclone frequency in certain regions of the North Atlantic Sector and for certain prediction periods, even for ensembles of ten or fewer members. While Kruschke et al. (2016) used a probabilistic approach to categorize cyclone frequency into tercile-based categories, they did not explicitly assess the skill of the model for each category separately. Haas

et al. (2015) found significant skill in MPI-ESM for upper quantiles of wind speeds at lead times of 1-4 years, but also noted that the skill decreases with lead time and is lower over the North Sea than over the adjacent land areas of Denmark, Germany, and the Netherlands. Moemken et al. (2021) confirmed the capability of a dynamically downscaled component of the MiKlip prediction system for additional wind-related variables, such as winter season wind speed and a simplified winter season storm severity index (e.g. Pinto et al., 2012). However, Moemken et al. (2021) noted that wind-based indices are usually less

skillful than variables based on temperature or precipitation, and are also heavily lead-time dependent (Reyers et al., 2019). Furthermore, the prediction skill of wind-based indices shows strong spatial variability, which prevents any generalization of the current state of prediction capabilities for regionally confined climate extremes.

In addition to the high variability of the decadal prediction skill for wind-based indices, the depiction of near-surface wind in

models strongly depends on the selected parameterization. Therefore, we circumvent the use of a wind-based index for evaluating the prediction skill for regional storm activity, and focus on a proxy that is based on horizontal differences of mean sea-level pressure (MSLP) and the resulting mean geostrophic wind speed instead. The index was first proposed by Schmidt and von Storch (1993) to avoid the use of long-term wind speed records, which oftentimes show inhomogeneities due to changes in the surroundings of the measurement site, and has already been used to reconstruct historical storm activity in the German Bight

(e.g. Schmidt and von Storch, 1993; Krieger et al., 2020). The geostrophic storm activity index is based on the assumption that the statistics of the geostrophic wind represent the statistics of the near-surface wind, which was confirmed by Krueger and von Storch (2011). The validity of the assumption is especially given over flat surfaces, like the open sea, where disturbances from friction are negligible. We therefore draw on the finding that the geostrophic wind-based index represents a suitable proxy for near-surface storm activity and can be used to derive some of the most relevant statistics of storm activity in the German

Bight. Furthermore, the index is particularly well suited for small regions, since calculating the MSLP gradient over a small area allows for the detection of small-scale variability of the pressure field, which is crucial for estimating geostrophic wind statistics.

Besides the choice of variables, the ensemble size also plays an important role in decadal prediction systems. The experi-
ments performed in MiKlip consisted of up to 10 members in the first two model generations, and 30 members in the third generation (Marotzke et al., 2016). Sienz et al. (2016) showed that larger ensembles generally result in better predictability, especially in areas with low signal-to-noise ratios. However, Sienz et al. (2016) also noted the number of ensemble members alone does not compensate for other potential shortcomings of the model. In a more recent study, Athanasiadis et al. (2020) found that larger ensemble sizes increase the decadal prediction skill for the North Atlantic Oscillation and high-latitude block-
ing. Furthermore, the use of a large ensemble increases the reliability of probabilistic predictions. The concept of a probabilistic approach is the presumption that a change in the shape of the ensemble distribution can be used to predict likelihoods of actual changes of climatic variables. In contrast to deterministic predictions, probabilistic predictions are also able to provide uncertainty information. With increasing ensemble size, and a resulting higher count of members in the tails of the predictive distribution, probabilistic predictions for extreme events, i.e. periods with very high or low storm activity, become feasible
(e.g., Richardson, 2001; Mullen and Buizza, 2002). Therefore, we build on these findings by increasing the ensemble size in this study to a total of 64 members.

In this study, we assess the prediction skill for GBSA of a 64-member ensemble of yearly initialized decadal hindcasts, i.e., forecasts for the past, based on the MPI-ESM. Since GBSA is connected to the large-scale circulation (Krieger et al., 2020),
we first analyze the ability of the decadal prediction system (DPS) to deterministically predict large-scale MSLP in the North Atlantic by comparing model ensemble mean output to data from the ERA5 reanalysis (Hersbach et al., 2020) (Sect. 3.1.1). In the German Bight, most of the annual storm activity can be attributed to the winter season. Therefore, we focus on the winter (December-February, DJF) mean MSLP and quantify the quality of deterministic predictions by correlating time series of predictions (ensemble mean) and observations. We show how positive correlations emerge in predictions of both winter MSLP
and GBSA (Sect. 3.1.2). We then evaluate the skill of the DPS for probabilistic predictions of MSLP and GBSA (Sect. 3.2.1 and 3.2.2), expressed via the Brier Skill Score ($BSS$, Brier, 1950), and discuss the advantages and limits of our approach (Sect. 3.3). Concluding remarks are given in Sect. 4.

## 2  Methods and Data

### 2.1  The Observational Reference

We use the time series of annual GBSA from Krieger et al. (2020) as an observational reference for the evaluation of prediction skill. The time series is based on standardized annual 95th percentiles of geostrophic wind speeds over the German Bight. The geostrophic winds are derived from triplets of three-hourly MSLP observations at eight measurement stations at or near

the North Sea coast in Germany, Denmark, and The Netherlands. MSLP measurements are provided by the International Surface Pressure Databank (ISPD) version 3 (Cram et al., 2015; Compo et al., 2015), as well as the national weather services of Germany (Deutscher Wetterdienst, DWD, 2019), Denmark (Danmarks Meteorologiske Institut, Cappelen et al., 2019), and the Netherlands (Koninklijk Nederlands Meteorologisch Instituut, KNMI, 2019). The time series of German Bight storm activity derived from observations covers the period 1897-2018.

Furthermore, we employ data from the ERA5 reanalysis (Hersbach et al., 2020), which has recently been extended backwards to 1950. The reanalysis data enables the prediction skill assessment over areas where in-situ observations are incomplete or too infrequent, for example over the North Atlantic Ocean.

## 2.2 MPI-ESM-LR Decadal Hindcasts

We investigate the decadal hindcasts of the MPI-ESM coupled climate model in version 1.2 (Mauritsen et al., 2019), run in low-resolution (LR) mode. The MPI-ESM-LR consists of coupled models for ocean and sea-ice (MPI-OM, Jungclaus et al., 2013), atmosphere (ECHAM6, Stevens et al., 2013), land surface (JSBACH, Reick et al., 2013; Schneck et al., 2013), and ocean biogeochemistry (HAMOCC, Ilyina et al., 2013). As we investigate the predictability of storm activity, which is derived from mean sea-level pressure, we focus on the atmospheric output given by the atmospheric component ECHAM6. The LR mode of ECHAM6 has a horizontal resolution of $1.875°$ (T63 grid), as well as 47 vertical levels between $0.1\,hPa$ and the surface (Stevens et al., 2013). The horizontal extent of the grid boxes is approximately $210\,km$ x $210\,km$ at the Equator, and $125\,km$ x $210\,km$ over the German Bight, which is still fine enough for the German Bight to cover multiple gridpoints. The model is forced by external radiative boundary conditions, which correspond to the historical CMIP6 forcing until 2014, and the SSP2-4.5 scenario starting in 2015 (contrary to CMIP5 and the RCP4.5 scenario used in the MiKlip experiments).

The ensemble members are initialized every November 1st from 1960 to 2019. The initialization and ensemble generation scheme is based on a system developed and tested within MiKlip (the "EnKF" system in Polkova et al. (2019)). For our study it has been updated from CMIP5 to CMIP6 external forcing, and extended from 16 to 80 ensemble members. The basis of this scheme is formed by a 16-member ensemble assimilation, which from 1958 to 2019 assimilates the observed oceanic and atmospheric state into the model (Brune and Baehr, 2020). In particular, an oceanic Ensemble Kalman filter is used with an implementation of the Parallel Data Assimilation Framework (Nerger and Hiller, 2013), and atmospheric nudging is applied. All 80 ensemble members of the predictions are directly initialized from the 16-ensemble member assimilation, with five different perturbations applied to the horizontal diffusion coefficient in the upper stratosphere to generate the total amount of 5x16=80 ensemble members. For example, hindcast members 1, 17, 33, 49, 65 are all initialized from assimilation member 1, but with different perturbation in the upper stratosphere (no perturbation for member 1, four different non-zero perturbations for the other members). Since we require three-hourly output (see Sect. 2.2.2), which is not available for the first 16 members of the 80-member ensemble, we constrict our analysis to the remaining 64 members. In the following, we will refer to these members as members 1-64. Due to the observational time series of German Bight storm activity from Krieger et al. (2020)

ending in 2018, we only evaluate hindcast predictions until 2018. For example, the last run considered in the evaluation for lead year 10 predictions is the one initialized in 2008, whereas the lead year 1 evaluation takes all runs initialized until 2017 into account.

### 2.2.1  Definition of Lead Times

All hindcast runs are integrated for 10 years and 2 months, each covering a time span from November of the initialization year (lead year 0) to December of the tenth following year (lead year 10). For consistency, we only consider full calendar years for the comparison, leaving us with ten complete years per intialization year and ensemble member. The ten individual prediction years are hereinafter defined as lead year $i$, with $i$ denoting the difference in calendar years between the prediction and the initialization. By this definition, lead year 1 covers months 3-14 of each integration, lead year 2 covers months 15-26, and so on. Lead year ranges are defined as time averages of multiple subsequent lead years $i$ through $j$ within a model run, and are called lead years $i$-$j$ in this study. To compare hindcast predictions for certain lead year ranges to observations, we average annual observations over the same time period (see Supplementary Material for more details).

It should be noted that winter (DJF) means are always labeled by the year that contains the months of January and February. A DJF prediction for lead year 4 therefore contains the December from lead year 3 plus the January and February from lead year 4. Likewise, a DJF prediction for lead years 4-10 contains every December from lead years 3 through 9, as well as every January and February from lead years 4 through 10.

In this study, we aim at drawing general conclusions about the prediction skill for North Atlantic MSLP anomalies for long and short averaging periods. Therefore, we focus on lead years 4-10, as well as lead year 7, as examples for long and short averaging periods for the prediction skill for MSLP anomalies, respectively. The choice of lead years 4-10 is based on selecting a sufficiently long averaging period that is representative of the characteristics of multi-year averages. Lead year 7 is chosen as it marks the center year within the lead year 4-10 period. We would like to note that the choice of lead years 4-10 and 7 is arbitrary, but we also analyse other comparable lead year periods (e.g., 2-8 and 5) to ensure sufficient robustness of our conclusions. However, we refrain from explicitly showing results for every lead time for reasons of brevity. For German Bight storm activity, which does not contain spatial information, we show the skill for all combinations of lead year ranges.

### 2.2.2  Geostrophic Wind and German Bight Storm Activity

For our analysis, we use three-hourly MSLP over the North Atlantic basin, including the German Bight. As three-hourly MSLP is only available as an output variable for the ensemble members 33-64, but not for 1-32, we use surface pressure $p$, surface geopotential $\Phi$ and surface temperature $T$ output from the model and apply a height correction. Following Alexandersson et al. (1998) and Krueger et al. (2019), the equation for the reduction of $p$ to the MSLP $p_0$ reads

$$p_0 = p \cdot \left( 1 - \frac{\Gamma \frac{\Phi}{g}}{T} \right)^{\frac{M \cdot g}{R \cdot \Gamma}} , \tag{1}$$


with the Earth's gravitational acceleration $g = 9.80665 \, \mathrm{m \, s^{-2}}$, the assumed wet-adiabatic lapse rate $\Gamma = 0.0065 \, \mathrm{K \, m^{-1}}$, the molar mass of air $M = 28.9647 \, \mathrm{g \, mol^{-1}}$, and the gas constant of air $R = 8.3145 \, \mathrm{J \, mol^{-1} \, K°{-}1}$. A consistency check between ensemble members 1-32 (manually reduced to sea level) and 33-64 (MSLP available as model output) resulted in negligible differences in MSLP (not shown). Therefore, we assume that the pressure reduction does not significantly influence our results

and treat the entire 64 member ensemble as a homogeneous entity.

We generate time series of German Bight storm activity (GBSA) in the MPI-ESM-LR hindcast runs. Owing to the low resolution of the model, we choose the three closest gridpoints that span a triangle encompassing the German Bight (Fig. 1). The coordinates of the selected gridpoints are specified in Table 1. The gridpoints are selected so that the resulting triangle is

sufficiently close to an equilateral triangle. This requirement is necessary to avoid a large error propagation of pressure uncertainties, which would cause a shift of the wind direction towards the main axis of the triangle (Krieger et al., 2020). We use three-hourly MSLP data from the decadal hindcast ensemble at the three corner points of the triangle and derive geostrophic winds from the MSLP gradient on a plane through these three points, following Alexandersson et al. (1998).

GBSA is defined as the standardized annual 95th percentiles of three-hourly geostrophic wind speeds. For each combination of ensemble member, initialization year, and forecast lead year, we determine the 95th percentile of geostrophic wind speed (exemplarily shown for one combination in Fig. 2). The percentile-based approach incorporates both the number and the strength of storms, thereby ensuring that both years with many weaker storms and years with fewer but stronger storms are represented as high-activity years. However, the proxy is not able to differentiate whether high storm activity is caused by

a large number of storms or by their high wind speed. The annual 95th percentiles of geostrophic wind speed take on values between 18 and $29 \, \mathrm{m \, s^{-1}}$ with an average of $22.87 \, \mathrm{m \, s^{-1}}$ (Fig. 3), which is close to the observational average of $22.19 \, \mathrm{m \, s^{-1}}$ derived by Krieger et al. (2020) for the period 1897-2018.

We accomplish the standardization by first calculating the mean and standard deviation of annual 95th percentiles of

geostrophic wind speeds from the runs initialized in 1960-2009 for lead year 1 and each member. We then subtract the means from the annual 95th percentiles, and divide by the standard deviations. Since the lead year 1 predictions started in 1960-2009 cover the period of 1961-2010, our standardization period matches the reference time frame used for storm activity calculation in Krieger et al. (2020). The resulting time series of lead year 4-10 and 7 ensemble mean predictions of GBSA, as well as the corresponding time series of observed GBSA are shown exemplarily in Fig. A1.


**Table 1.** Coordinates of the three gridpoints used for storm activity calculation in the model.

| Gridpoint | Latitude ($^\circ$ N) | Longitude ($^\circ$ E) |
|-----------|-----------------------|------------------------|
| North     | 55.02                 | 9.38                   |
| West      | 53.16                 | 5.63                   |
| Southeast | 53.16                 | 9.38                   |

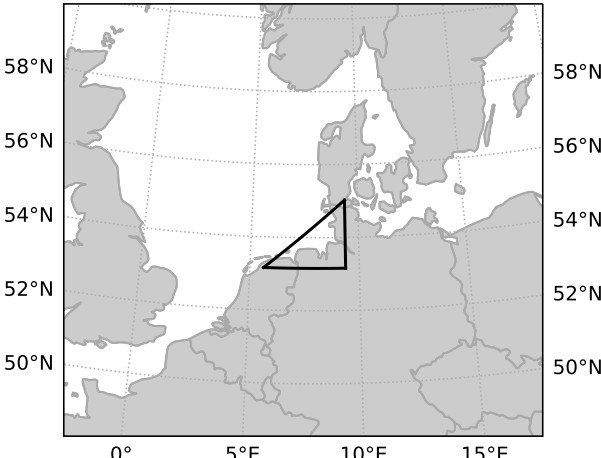

**Figure 1.** Map of Northwestern Europe, showing the location of the German Bight triangle.

While the analysis of GBSA only uses MSLP data from three gridpoints in the German Bight, we also analyse the prediction skill for MSLP anomalies over the entire North Atlantic.

## 2.3 Evaluation of Model Performance

In this study, we evaluate the model's performance for both deterministic and probabilistic predictions. First, we evaluate deterministic predictions to quantify the ability of the model to capture the variability of GBSA. Second, we analyze probabilistic predictions to examine whether the large ensemble is able to skillfully differentiate between extremes and non-extremes. These two prediction types require different evaluation metrics.

### 2.3.1 Anomaly Correlation

For deterministic predictions, we calculate Pearson's anomaly correlation coefficient ($ACC$) between predicted and observed quantities:

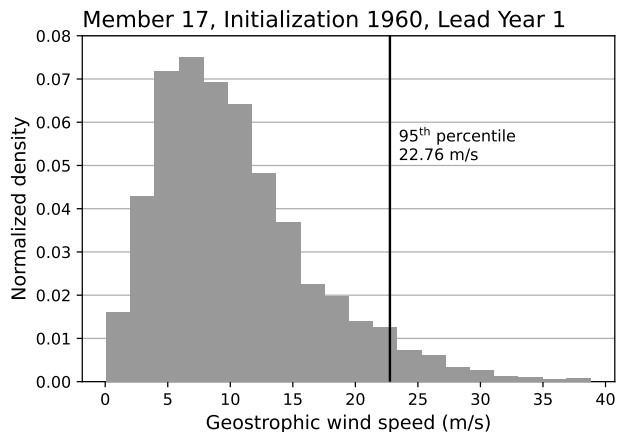

**Figure 2.** Exemplary distribution of predicted three-hourly geostrophic wind speeds for lead year 1 from member 17, initialized in 1960. The vertical line marks the 95th percentile, which is used in the calculation of storm activity.

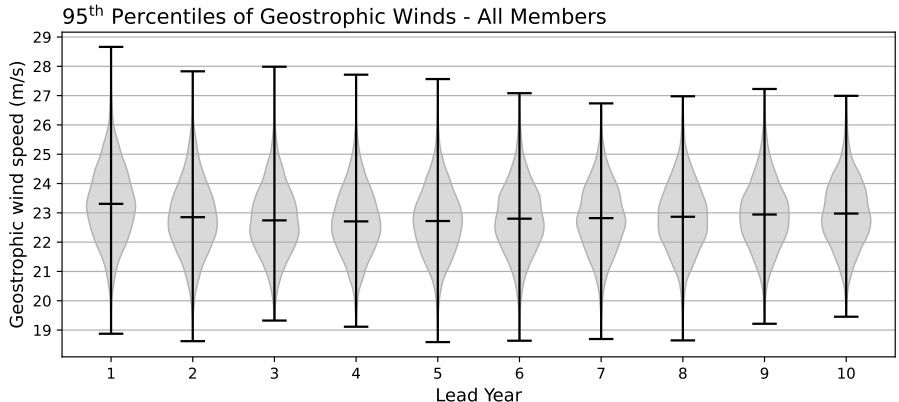

**Figure 3.** Violin plot of the distribution of annual 95th percentiles of geostrophic wind speeds from all members and all initializations, separated by lead year. Lead years increase from left to right along the x-axis. The width of the violin indicates the normalized density for a certain wind speed. Horizontal dashes mark maxima, means and minima for each lead year.

$$ACC = \frac{\sum_{i=1}^{N}(f_i - \bar{f})(o_i - \bar{o})}{\sqrt{\sum_{i=1}^{N}(f_i - \bar{f})^2 \sum_{i=1}^{N}(o_i - \bar{o})^2}}, \tag{2}$$

with the predicted and observed quantities $f_i$ and $o_i$, as well as the long-term averages of predictions and observations $\bar{f}$ and $\bar{o}$. The $ACC$ can take on values from 1 to -1, with 1 indicating a perfect correlation, 0 equating to no correlation, and -1 showing a perfect anticorrelation. The statistical significance of the $ACC$ is determined through a 1000-fold moving block

bootstrapping with replacement (Kunsch, 1989; Liu, 1992), where the 0.025 and 0.975 quantiles of bootstrapped correlations define the range of the 95 % confidence interval. The block length is set to $k = 4$, following the suggestion of $k = \mathcal{O}(N^{\frac{1}{3}})$ (Lahiri, 2003) for a number of datapoints $N$ between 50 and 60, depending on the variable and the length of the averaging period. The mean $ACC$ is calculated by applying a Fisher z-transformation (Fisher, 1915) to the bootstrapped correlations, averaging over all values in z-space, and transforming the average back to the original space. The transformation of correlations $ACC$ to z-scores $z$ and its inverse are defined as $z = \operatorname{arctanh}(ACC)$ and $ACC = \tanh(z)$, where $\tanh$ and $\operatorname{arctanh}$ are the hyperbolic tangent function and its inverse, respectively.

### 2.3.2 Brier Skill Score

Probabilistic predictions are evaluated against a reference prediction (see Sect. 2.5) by employing the strictly proper Brier Skill Score ($BSS$, Brier, 1950). The $BSS$ is a skill metric for dichotomous predictions and is defined as

$$BSS = 1 - \frac{BS}{BS_{\mathrm{ref}}}, \tag{3}$$

where $BS$ and $BS_{\mathrm{ref}}$ denote the Brier Scores of the probabilistic model prediction and a reference prediction, respectively. This definition results in positive $BSS$ values whenever the model performs better than the chosen reference, and negative values when the reference outperforms the model. A perfect prediction would score a $BSS$ of 1. The statistical significance of the $BSS$ is calculated through a 1000-fold bootstrapping with replacement. We perform the bootstrapping in temporal space by selecting random blocks with replacement, but do not bootstrap across the ensemble space. In this study, we use a significance level of 5 % to test whether model performance is significantly different from the reference.

The Brier Score $BS$ is defined as

$$BS = \frac{1}{N} \sum_{i=1}^{N} (F_i - O_i)^2, \tag{4}$$

with the number of predictions $N$, the predicted probability of an event $F_i$ and the event occurrence $O_i$. The predicted probability $F_i$ is determined by the number of ensemble members that predict the event divided by the total ensemble size of 64. Note that $O_i$ always takes on a value of either 1 or 0, depending on whether the event happened or not. Because the $BS$ is calculated as the normalized mean square error in the probability space, it is negatively oriented with a range of 0 to 1, i.e.,

better predictions score lower $BS$ values. A prediction based on flipping a two-sided coin ($F_i = 0.5$) would score a $BS$ of 0.25.

We are interested in the skill of probabilistic predictions of periods of high, moderate, and low storm activity, as well as high, moderate, and low winter MSLP anomalies. To differentiate between events and non-events, the $BS$ needs thresholds, which we set to 1 and -1. We define high activity periods as time steps above 1, low activity periods as time steps below -1, and moderate activity periods as the remaining time steps. Since the $BSS$ can only assess the skill of dichotomous predictions, we evaluate each of the three respective categories (high, moderate, low) separately. This methodology differs from Kruschke et al. (2016), as we do not evaluate one three-category forecast, but three two-category forecasts instead.

## 2.4   Re-standardization of Multi-year Averages

Winter MSLP anomalies and GBSA time series are standardized before the analysis. To keep the evaluation of multi-year averaging periods consistent with that of single lead years, we re-standardize all time series after applying the moving average. We do this since the thresholds of our probabilistic prediction categories require the underlying data to be normally distributed with a mean of 0 and a standard deviation of 1 by definition. For spatial fields, we perform the standardizations and skill calculations gridpoint-wise. As GBSA is based on the mean MSLP gradient of a plane through three gridpoints, we treat its spatial information like that of a single gridpoint and calculate skill metrics only once for the entire plane.

## 2.5   Reference Forecasts

The $BSS$ evaluates the skill of probabilistic predictions against a reference prediction. In this study, we use both a deterministic persistence prediction and a probabilistic climatological random prediction as a baseline against which we test the prediction skill of the MPI-ESM-LR, which is a common practice in climate model evaluation (e.g. Murphy, 1992).

The deterministic persistence prediction of storm activity is generated by taking the average observed storm activity of $n$ years before the initialization year of the model run. $n$ is defined to be equal to the length of the predicted lead year range. For example, a lead year 4-10 prediction ($n = 7$) initialized in 1980 is compared to the persistence prediction based on the observed average of the years 1973-1979, whereas a lead year 7 prediction ($n = 1$) from the same initialization is compared to the persistence prediction based on the observed storm activity of 1979. Persistence predictions of winter MSLP are generated likewise but use ERA5 reanalysis data instead of direct observations. We note that since the persistence prediction is not probabilistic, it can either be correct or incorrect in a given year, which corresponds to the term $(F_i - O_i)$ in Eq. 4 taking on a value of either 0 (correct) or 1 (incorrect).

The probabilistic climatological random prediction uses the climatological frequencies of observed events (e.g. Wilks, 2011). As our time series of winter MSLP anomalies and GBSA are normally distributed by definition, the climatological frequencies can be derived from the Gaussian normal distribution. For instance, a climatological random prediction for high storm activity, which is defined via a threshold of one standard deviation above the mean, would always predict a fixed occurrence probability

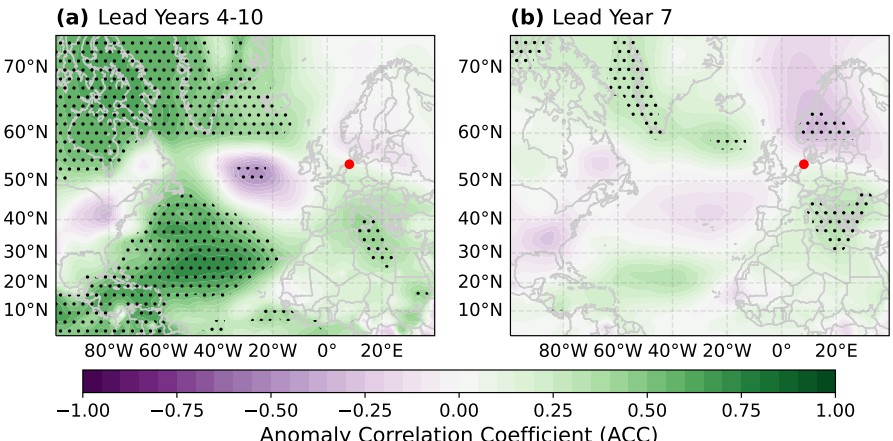

**Figure 4.** Gridpoint-wise anomaly correlation coefficient ($ACC$) between the deterministic hindcast ensemble mean prediction of winter mean (DJF) MSLP anomalies and ERA5 reanalysis data for lead years 4-10 **(a)** and lead year 7 **(b)**. The German Bight is marked by a red dot. Anomalies are calculated for each member individually and averaged over the entire ensemble afterwards. Stippling indicates significant correlations ($p \leq 0.05$).

of $F_i = 1 - \Phi(1) = 0.1587$. Here, $\Phi(x)$ describes the cumulative distribution function of the normal distribution. $\Phi(x)$ gives the probability that a sample drawn from the Gaussian normal distribution at random is smaller or equal to $\mu + x\sigma$, with $\mu$ and $\sigma$ denoting the mean and standard deviation of the distribution, respectively.

## 3  Results and Discussion

### 3.1  Deterministic Predictions

#### 3.1.1  Mean Sea-Level Pressure

Since geostrophic storm activity is an MSLP-based index, we first investigate the correlation between the model's deterministic predictions of winter (DJF) MSLP and data from the ERA5 reanalysis product, expressed as the gridpoint-wise anomaly correlation coefficient ($ACC$). For lead year 4-10 winter MSLP anomalies, the $ACC$s are positive over larger parts of the subtropical Atlantic, as well as Northeastern Canada and Greenland (Fig. 4a). Negative $ACC$s emerge in a circular area west

of the British Isles. Over the German Bight, however, the $ACC$ for winter MSLP anomalies is insignificant. The pattern over the subtropical Atlantic Ocean agrees with the multi-model study by Smith et al. (2019), who found significant skill for winter MSLP in similar regions at lead years 2-9. Smith et al. (2019) however also found skill over Scandinavia, where our DPS fails to provide any evidence of skill for long averaging periods. The $ACC$ pattern of lead year 4-10 is also present for most other lead year ranges with averaging periods of 5 or more years (not shown).


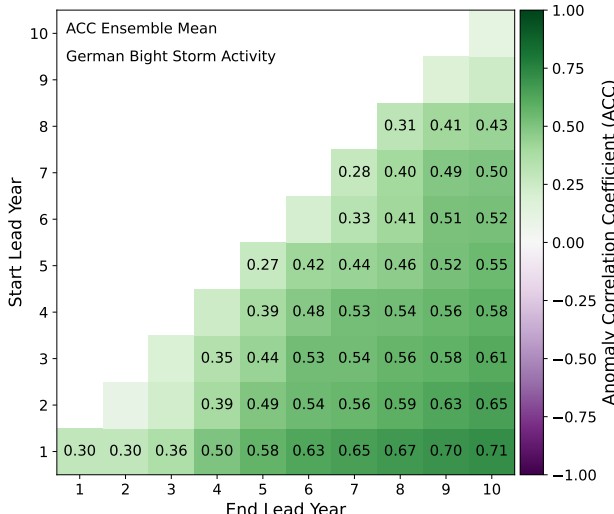

**Figure 5.** Anomaly correlation coefficients between the deterministic DPS forecasts and observations of German Bight storm activity for all combinations of start (y-axis) and end lead years (x-axis). Numbers in boxes indicate those correlation coefficients that are significantly different from 0 ($p \leq 0.05$).

For the single lead year 7, the $ACC$ is negative over Scandinavia. Across the rest of the spatial domain, the absolute values of the $ACC$ are lower for lead year 7 (Fig. 4b) than for lead year 4-10, but the pattern shows some similarity. Again, the $ACC$ is insignificant over the German Bight, indicating an insufficient skill to properly predict winter MSLP anomalies. The characteristics of the $ACC$ distribution in Fig. 4b also hold for other single lead years, suggesting that longer averaging periods generally result in higher absolute correlations, both for regions with positive and negative correlation values.

### 3.1.2 Storm Activity

We find that the $ACC$ between ERA5 and DPS predictions for winter MSLP is significantly positive in certain regions of the North Atlantic, especially when averaged over multiple prediction years, but falls short of being significant over the German Bight. Still, the general predictive capabilities of the DPS for winter MSLP motivates the investigation of GBSA predictability. Fig. 5 shows the deterministic predictability of GBSA, expressed as the $ACC$ between the model ensemble mean and observations for all possible lead time combinations. Here, single lead years are displayed along the diagonal, while the length of the averaging period increases towards the bottom right corner. The $ACC$ for GBSA is insignificant for most single prediction years (except for lead years 1, 5, 7, and 8), but it increases towards longer averaging periods. The $ACC$ exhibits a clear dependence on the length of the averaging period, with lead years 1-10 showing the highest overall $ACC$ among all lead year ranges ($r = 0.71$). Apart from lead years 2-3 and 9-10, the ensemble mean tends to become more skillful with longer averaging periods, and shows significant positive $ACC$s for all multi-year prediction periods. This stands in clear contrast to the results for winter MSLP predictions, where the model failed to produce significant $ACC$s for both short and long averaging periods

in the German Bight (compare Fig. 3.1.1).

Similar to the predictability of winter MSLP (Sect. 3.1.1), we find a dependency of GBSA predictability on the length of the averaging window. Again, we argue that this may be caused by smoothing out the short-term variability that is apparent in reconstructed time series of annual GBSA (Krieger et al., 2020). However, the $ACC$ is notably independent on the lead time. We would expect a deterioration of the $ACC$ with increasing temporal distance from the initialisation, i.e. along the diagonal in Fig. 5. Instead, we observe a relative hotspot of predictability for lead year ranges of 2 to 4 years that start at lead year 3 and

4 (i.e., lead years 3-4 till 3-6 and 4-5 till 4-7). These ranges demonstrate higher predictability than comparable ranges closer to the present.

## 3.2    Probabilistic Predictions

Since the deterministic predictions investigated so far are based on the ensemble mean, they do not take the ensemble spread into account. Therefore, we now make use of the large ensemble size to also generate probabilistic predictions for high,

moderate, and low storm activity events, as well as high, moderate, and low winter MSLP anomaly events. We expect the DPS to be skillful in predicting probabilities since the large ensemble size allows us to detect changes in the shape of the ensemble distribution.

### 3.2.1    Mean Sea-Level Pressure

When predicting positive winter MSLP anomalies (Fig. 6a and 6b), the DPS significantly outperforms persistence ($BSS > 0$)

over large parts of the Central North Atlantic and Europe for both lead years 4-10 and 7. Over the North Sea, however, the $BSS$ of the model is indistinguishable from 0 for lead years 4-10, indicating very limited skill to correctly predict positive winter MSLP anomalies. For lead year 7 predictions of positive winter MSLP anomalies, the $BSS$ is slightly higher over the North Sea, with a higher model skill than that of persistence for most of the gridpoints. A similar pattern is found in predictions of negative anomalies (Fig. 6c and 6d), where the DPS does not show any additional skill compared to persistence over the

North Sea for lead years 4-10, but improves for lead year 7. Most notably, the DPS outperforms persistence in the far North Atlantic for lead years 4-10, but fails to do so in the subtropical North Atlantic.

Predictions of moderate winter MSLP anomalies (Fig. 6e and 6f) are skillful compared to persistence over most of the spatial domain. Still, a region of poor skill emerges over the German Bight and adjacent areas for lead year 4-10 predictions, while

lead year 7 predictions show a $BSS$ significantly higher than 0. The high $BSS$ values of moderate anomaly predictions, however, are caused by poor performance of the persistence prediction serving as a reference. The $BS$ of this reference prediction is significantly higher than 0.25 (not shown), demonstrating that persistence predictions are less skillful than a coin flip-based prediction which assumes an occurrence probability of 50 % for every year. Hence, the $BSS$ against persistence alone should not be used to infer the skill of the DPS for winter MSLP anomaly events.


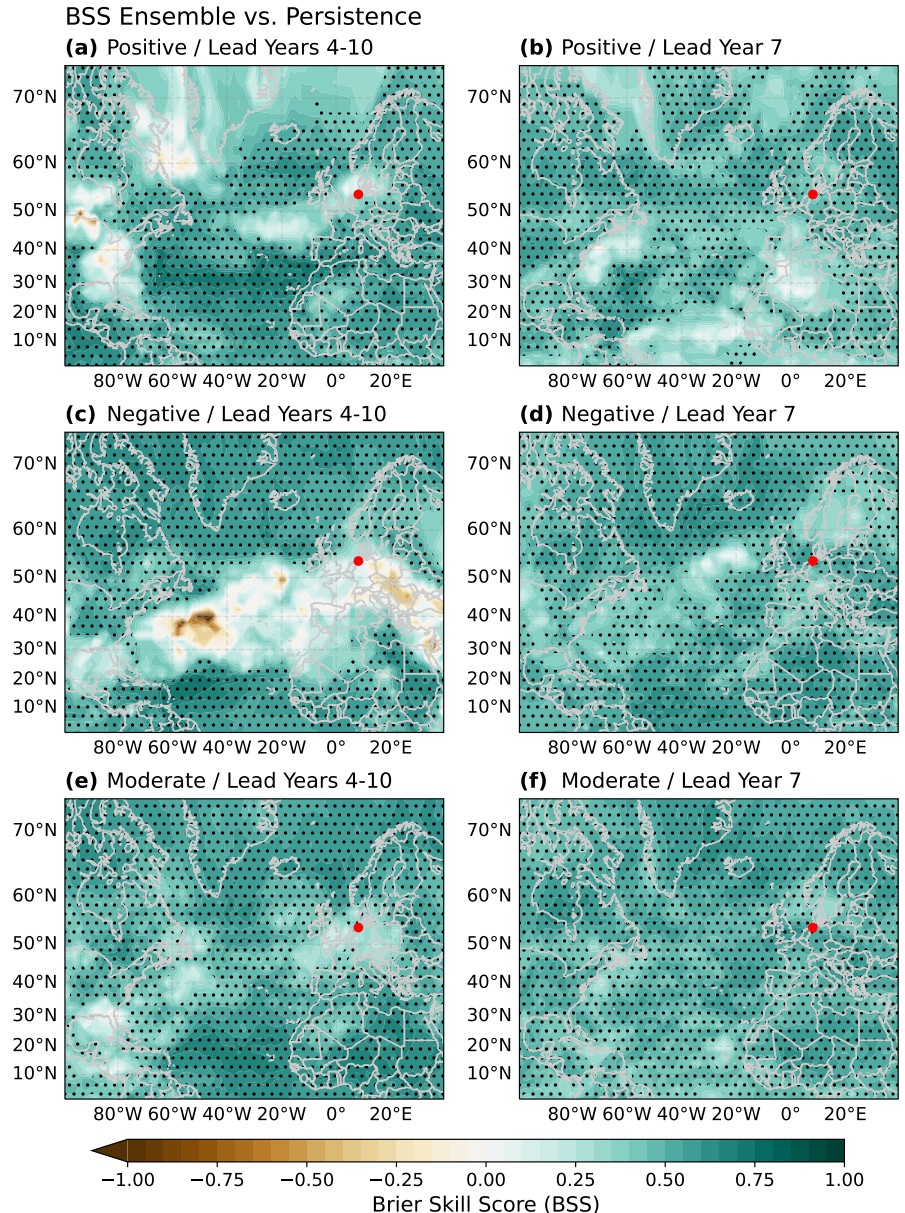

**Figure 6.** Prediction skill of probabilistic forecasts of positive **(a,b)**, negative **(c,d)**, and moderate **(e,f)** winter mean (DJF) MSLP anomalies, expressed as the Brier Skill Score ($BSS$) of the 64 member ensemble evaluated against a persistence prediction as a baseline for lead years 4-10 **(a,c,e)** and lead year 7 **(b,d,f)**. Thresholds for event detection are set to -1 and 1. The German Bight is marked by a red dot. Stippling marks areas with a $BSS$ significantly different from 0 ($p \leq 0.05$).

Therefore, we additionally test the skill of the model for winter MSLP anomalies against that of a climatology-based prediction (Fig. 7). The model $BSS$ compared to climatology is mostly indistinguishable from 0 for both lead years 4-10 (Fig. 7a,

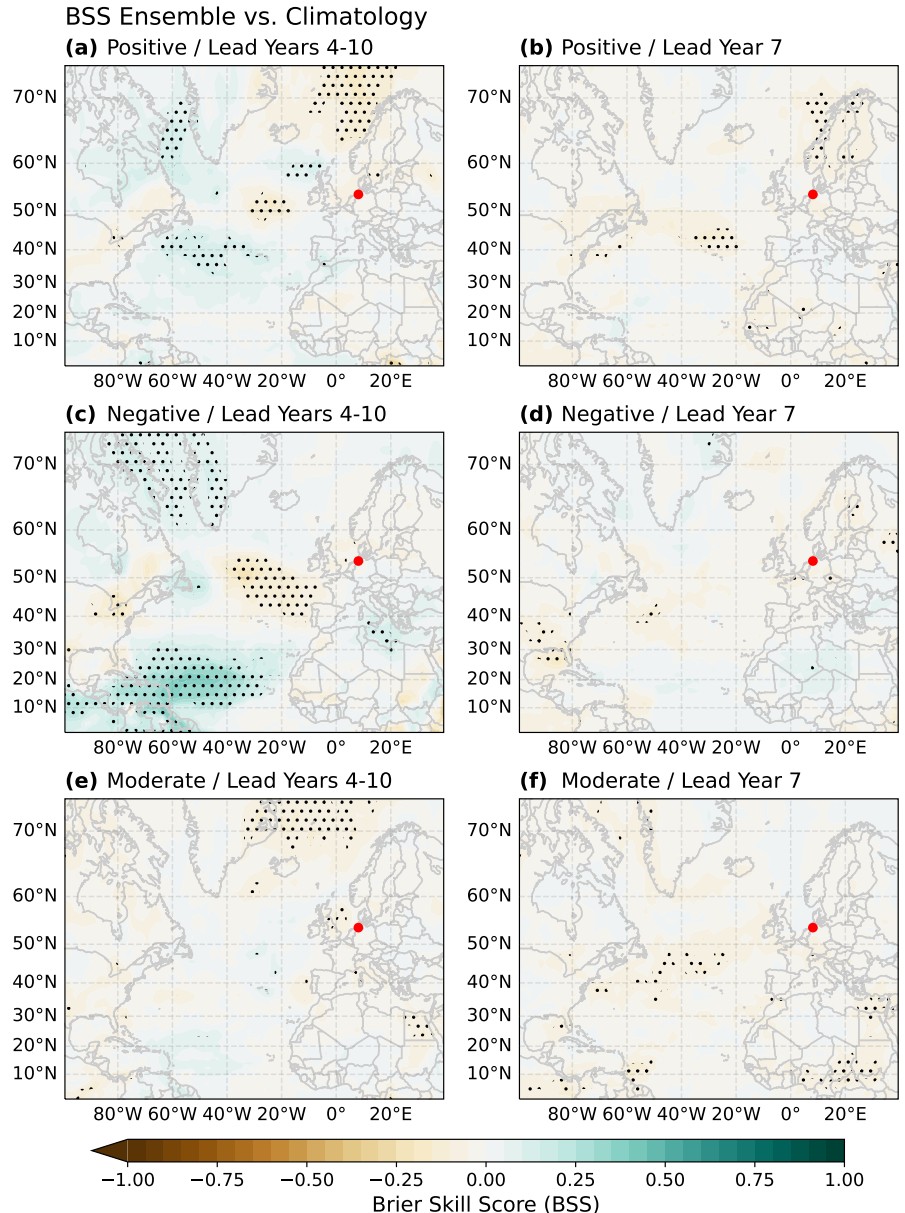

**Figure 7.** Like Fig. 6, but evaluated against a climatology-based prediction as a baseline.

7c, and 7e) and 7 (Fig. 7b, 7d, and 7f), indicating a very limited potential of the DPS to outperform climatology over vast parts of the North Atlantic sector. Large patches of positive $BSS$ values are found in lead year 4-10 predictions of negative winter MSLP anomalies over the tropical Atlantic (Fig. 7c), whereas negative $BSS$ values emerge over the polar North Atlantic for lead year 4-10 predictions of positive and moderate winter MSLP anomalies (Fig. 7a and 7e), as well as over the central North

Atlantic for lead year 4-10 predictions of negative winter MSLP anomalies (Fig. 7c).

Overall, the DPS appears to predict positive and negative German Bight winter MSLP anomalies better than persistence for short averaging periods, while it fails to significantly outperform persistence for longer averaging periods. In addition, the DPS fails to consistently outperform climatology over large parts of the North Atlantic region for both short (lead year 7) and long (lead year 4-10) averaging periods. The comparison to climatology indicates that the high skill of the model when tested against persistence is caused by poor performance of the persistence prediction, rather than the prediction quality of the model. Nevertheless, the model shows some potential to bring additional value to the decadal predictability of winter MSLP anomalies.

### 3.2.2 Storm Activity

The skill evaluation of probabilistic winter MSLP predictions shows that the $BSS$ of the DPS for positive and negative anomalies are significantly better than those of persistence for large parts of the spatial domain. However, for long averaging periods, we do not observe a significant difference in skill between the DPS and persistence over the German Bight. Also, the model fails to outperform climatology for most parts of the North Atlantic sector. We now investigate the skill of probabilistic predictions of high, moderate, and low storm activity events, again using persistence and climatology as our baselines.

For high storm activity predictions, the $BSS$ against persistence is positive for all lead year combinations, indicating a better performance of the DPS than persistence (Fig. 8a). The $BSS$ is significantly positive for most 1-2 year averaging windows, as well as for very long averaging windows (7 years or more). When testing the model's high storm activity predictions against a climatology-based forecast (Fig. 8b), we find that the model exhibits significant skill for most averaging periods with a length of 4 or more years, but shows no skill for short averaging periods. The distribution of significant $BSS$ values among the lead year combinations against climatology differs strongly from the one obtained through testing against persistence (compare Fig. 8a), and much rather resembles the distribution of anomaly correlation coefficients between the deterministic predictions and observations (see Fig. 5). Furthermore, the $BSS$ against climatology is lower than against persistence for most lead year periods, indicating that climatology generally poses a tougher challenge for the model than persistence.

For low storm activity prediction (Fig. 8c), the $BSS$ is again positive for all lead year combinations. The $BSS$ is significantly different from 0 for single year and 3-year range predictions except for lead year 2, and lowest for averaging periods of 5-7 years. The higher $BSS$ for single years than for periods of 5-7 years indicates that the model is valuable at predicting short periods. This behavior agrees with the findings in Sect. 3.2, which significantly demonstrated positive skill for German Bight winter MSLP anomalies for a short period (lead year 7), but not for a multi-year average (lead years 4-10). However, the model only outperforms climatology (Fig. 8d) for lead year 3-10, while all other lead years show insignificant $BSS$ values. This suggests that while the model is able to beat a persistence-based prediction, it does not present any additional skill compared

 to climatology.

Moderate storm activity predictions (Fig. 8e) also exhibit positive $BSS$ values for all lead year ranges compared to persistence, and are significantly different from 0 except for lead years 8-9. However, this apparent high skill compared to persistence is once again only caused by the relative underperformance of the persistence prediction. A comparison with climatology (Fig. 8f) confirms that the model significantly outperforms climatology for lead year 2-3 only, and shows a reduced skill for lead years 5, 5-6, and 10, while it does not differ in skill for all remaining lead years.

Overall, the skill of the probabilistic forecast mostly depends on the choice of reference. While the model outperforms persistence over the majority of lead times in all three categories (high, moderate, low), it only outperforms climatology in predicting high storm activity for longer averaging windows. For probabilistic predictions of moderate and low storm activity, the model does not outperform climatology. Predictions of high storm activity with an averaging window of 6 or more years are the only ones where the model outperforms both climatology and persistence.

### 3.3 Discussion

We find that the $ACC$ between deterministic predictions and observations of winter MSLP anomalies over large parts of the North Atlantic and GBSA is positive and significantly different from 0 for most multi-year averaging periods. Over the German Bight, however, $ACC$s for winter MSLP anomaly predictions are insignificant. We hypothesize that while the model is unable to deterministically predict winter MSLP anomalies over the German Bight, it is able to predict the annual upper percentiles of MSLP gradients sufficiently well for the $ACC$s of GBSA to become significant. This might be due to the model showing some predictive capabilities for sufficiently large deviations from the mean, but not for fluctuations around the mean.

The general lead-year dependence of the magnitude of the $ACC$ agrees with previous findings of Kruschke et al. (2014), Kruschke et al. (2016), and Moemken et al. (2021) for other storm activity-related variables. In our study, the correlation between reanalysis and prediction mainly depends on the length of the lead time window, rather than the lead time (i.e., the temporal distance between the predicted point in time and the model initialization). We hypothesize that this dependency might be attributable to the filtering of high-frequency variability by the longer averaging windows, in combination with the model's ability to better predict the underlying low-frequency oscillation in the large-scale circulation. While our model is unable to deterministically predict the short-term variability within records of GBSA, these year-to-year fluctuations are smoothed out in predictions of multi-year averages, resulting in a higher $ACC$. Additionally, we would like to note that temporal autocorrelation might account for a part of these high $ACC$ values. Smoothing that results from the multi-year averaging process introduces dependence to the time series which may lead to artificially inflated $ACC$s compared to non-smoothed time series.

The lack of a dependency of the $ACC$ on the temporal distance from the initialization, however, cannot be explained by multi-year averaging. The relative hotspot of predictability for lead year ranges of 2 to 4 years starting at lead year 3 and 4 is counter-intuitive, especially due to the insignificant $ACC$s for lead years 2, 3, 4, and 2-3. These insignificant $ACC$s between

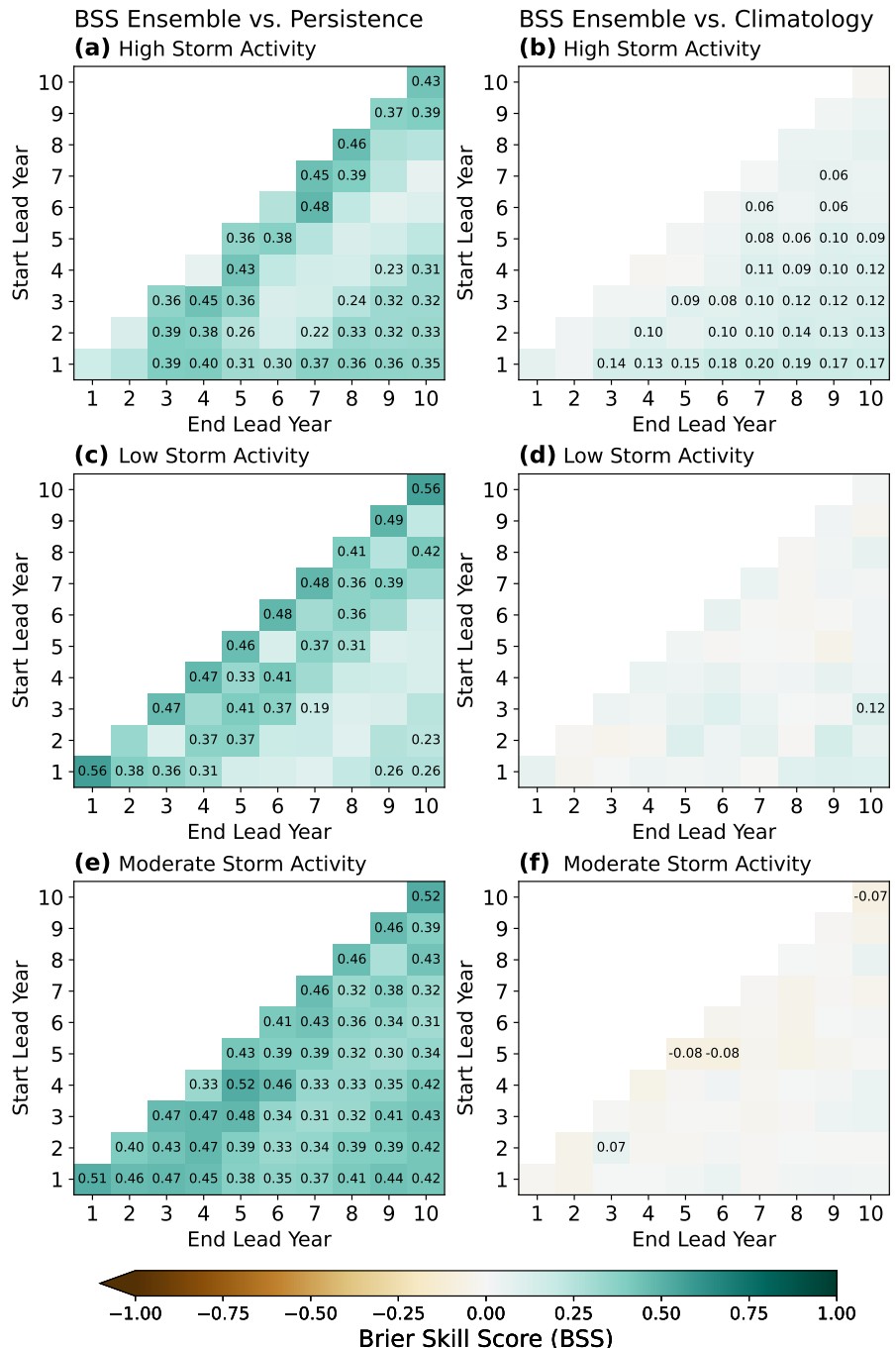

**Figure 8.** Brier Skill Score ($BSS$) of the 64 member ensemble for high **(a,b)**, low **(c,d)**, and moderate **(e,f)** storm activity evaluated against both a persistence-based **(a,c,e)** and a climatology-based **(b,d,f)** prediction as a baseline, shown for all combinations of start (y axis) and end lead years (x axis). Numbers in boxes are those $BSS$ that are significantly different from 0 ($p \leq 0.05$). Storm activity levels of 1 and -1 are used to differentiate between high, moderate, and low storm activity.

GBSA observations and deterministic predictions hint at a possible initialization shock influencing the model performance. In fact, the average geostrophic wind speed for lead years 2, 3, and 4 is lower than for lead year 1 (Fig. 3), supporting the hypothesis. Since all annual percentiles are standardized using lead year 1 as a reference, we expect the resulting standardized storm activity for lead years 2, 3, and 4 to be slightly lower than for lead year 1. However, the average geostrophic wind speeds for lead years 5 through 10 are also lower than for lead year 1, yet the $ACC$s for these lead years are significant again. In addition, we tested whether standardizing each lead year with its respective mean and standard deviation (instead of always using lead year 1) has a notable effect on the $ACC$. We find that the $ACC$ between model and observation is almost unaffected by the choice of our standardization reference (not shown). Hence, we rule out an initialization shock as the main reason for the low $ACC$s for lead years 2, 3, and 4. Beyond that, we are unable to come up with a convincing explanation for this behavior at this point. Thus, further studies are needed to investigate why the $ACC$ does not steadily decline with increasing lead times.

For probabilistic predictions, the choice of reference plays a crucial role in the evaluation of the DPS. Since we test the performance of the model against that of persistence- and climatology-based predictions, the $BSS$ not only depends on the prediction skill of the model, but also on the skill of the reference. Most likely, a significant $BSS$ is less a result of exceptional model performance, but rather indicates the limits of persistence. This dependence becomes overtly apparent during the analysis of moderate GBSA predictability. Moderate GBSA predictability is skillful when evaluated against a persistent reference prediction. However, this significant prediction skill turns mostly insignificant when evaluated against a climatology-based prediction. On the contrary, we also find certain lead times where high storm activity predictions by the DPS beat climatology, but fail to beat persistence.

The performance of persistence also contributes to the inverse dependency of the probabilistic skill on the length of the averaging window (i.e., a higher skill for shorter periods) that emerges in predictions of German Bight MSLP anomalies when tested against persistence. Here, the DPS exceeds the skill of persistence for short averaging periods, but fails to do so for long averaging periods. This contradicts the assumption of the capability of the DPS to skillfully predict the underlying low-frequency variability (see Sect. 3.1). However, the inverse dependency is more likely a result of better performance by the persistence prediction for longer averaging periods, which in turn challenges our model more than for short averaging periods. When evaluating probabilistic predictions of high GBSA against climatology, we find a similar dependency of the skill on the length of the averaging window as within deterministic predictions (i.e., a higher skill for longer periods), further confirming that the inverse dependency is an artifact of the performance of persistence.

Despite the aforementioned potential deficiencies, both persistence and climatology still range among the most appropriate references predictions to evaluate extreme GBSA predictability. We therefore conclude that our DPS is particularly valuable at lead times during which the reference forecasts are sufficiently poor. Vice-versa, the benefits of a DPS are negligible at lead times during which the skill of the reference forecast is sufficiently fair. Naturally, we cannot determine in advance which of the two reference predictions will be more skillful at predicting GBSA. For most lead year periods, however, climatology poses

a tougher challenge for the model than persistence, so we argue that outperforming climatology is an indication that the model
can bring added value to GBSA predictability.

The separation of the probabilistic predictions into three categories also demonstrates the necessity to evaluate the skill for each prediction category individually. By individually assessing the skill for each forecast category, we find that the model is more skillful than both persistence and climatology in predicting high storm activity periods for averaging windows longer
than 5 years. We emphasize that evaluating three separate two-category forecasts is not as challenging to the model as incorporating all three categories into one aggregated skill measure (e.g., the Ranked Probability Skill Score, or RPSS, Epstein, 1969; Murphy, 1969, 1971). Yet, our analysis allows us to detect that our model shows skill in regions where previous studies that used a combined probabilistic skill score did not find any skill for storm-related quantities (e.g., Kruschke et al., 2016), a conclusion which would have not been possible to draw by evaluating a single three-category prediction.

Our results for probabilistic predictions suggest that our approach of employing a large ensemble notably aids the model's prediction skill. Contrary to previous studies on the decadal predictability of wind-related quantities, we find significant skill for high storm activity in the German Bight, especially for long averaging periods, where model outperforms both persistence and climatology. The size of the ensemble might contribute to this skill, as similar analyses with smaller subsets of the DPS en-
semble resulted in a slightly lower prediction skill (not shown), confirming the findings of Sienz et al. (2016) and Athanasiadis et al. (2020). However, the impact on prediction skill by a further increase in the number of members is yet to be investigated.

As this study is based on a single earth system model, the inherent properties of the MPI-ESM-LR might impact our findings. Thus, our conclusions drawn from these findings are only valid for this model. Model intercomparison studies for the decadal
predictability of regional storm activity might eliminate the influence of possible model biases and errors. These intercomparisons will become possible once additional large-ensemble DPS products based on other earth system models are released.

It seems noteworthy that this study assumes annual storm activity and winter MSLP anomalies to be normally distributed, since the standardization process in the calculation of storm activity and winter MSLP anomalies fits a normal distribution to
the data. Other distributions (e.g., a Generalized Extreme Value distribution) might also be suited for a similar analysis, and could provide an additional opportunity to enhance the description of storm activity and, thus, further improve the probabilistic prediction skill in the future.

## 4   Summary and Conclusions

In this study, we evaluated the capabilities of a decadal prediction system (DPS) based on the MPI-ESM-LR to predict win-
ter MSLP anomalies over the North Atlantic region and German Bight storm activity (GBSA), both for deterministic and probabilistic predictions. The deterministic predictions are based on the ensemble mean, whereas the probabilistic predic-

tions evaluate the distribution of the 64 ensemble members. We assessed the anomaly correlation coefficient ($ACC$) between deterministic predictions and observations or reanalysis data, respectively, evaluated probabilistic predictions for three different forecast categories with the Brier Skill Score ($BSS$), and tested the probabilistic predictions of GBSA against both a
persistence- and a climatology-based prediction.

Through comparison with data from the ERA5 reanalysis, we found that the DPS produces poor deterministic predictions of winter MSLP anomalies over the German Bight. Over the North Atlantic, certain regions with higher correlations emerge, but the magnitude of the $ACC$ is heavily dependent on the length of the averaging window. In general, longer averaging periods
result in higher absolute correlations. The predictability for GBSA also depicts this same dependency on the averaging period, where $ACC$s are only significant for most averaging periods larger than 1 year.

Probabilistic predictions of winter MSLP anomalies over the North Atlantic are mostly skillful with respect to persistence, but do generally not show additional skill compared to climatology. For the German Bight in particular, only predictions for
short lead year ranges are skillful with respect to persistence, while predictions for longer averaging periods exhibit poor skill.

For probabilistic predictions of high storm activity, averaging windows of 6 or more years are more skillfully predicted by the DPS than by both persistence and climatology. This study demonstrates that the model does bring an improvement to predictability of GBSA, and that a separation into multiple prediction categories is essential to detecting hotspots of predictability
in the DPS which would have gone unnoticed in a more aggregated skill evaluation. Furthermore, we want to emphasize the ability of the DPS to especially issue reliable predictions for high storm activity, as this is arguably the most important category for which we could hope to achieve any prediction skill.

The high skill of probabilistic predictions for high storm activity, combined with the advantage of large-ensemble decadal
predictions, can be expected to bring benefits to stakeholders, operators and the society in affected areas by improving coastal management and adaptation strategies. By employing a large-ensemble DPS and carefully selecting a fitting prediction category, even regional climate extremes like GBSA can be skillfully predicted on multiannual to decadal timescales. With ongoing progress in the research field of decadal predictions, and advancements in model development, we are therefore confident that this approach opens up new possibilities for research and application, including the decadal prediction of other regional climate
extremes.

## Appendix A:  Comparison of Multi-Year Averages

In order to compare hindcast predictions for different lead year ranges to observations, we average hindcast predictions and observations over the same time periods. For example, a hindcast for lead years 4-10, which by definition is formed by averaging over a 7-year period, is always compared to a 7-year running mean of an observational dataset. The point-wise comparison of

time series is performed in such a way that the predicted time frame matches the observational time frame. In other words, the lead year 4-10 prediction from a run initialized in 1960, which covers the years 1964-1970, is compared to the observational mean of 1964-1970. To form a time series from the model runs, the predictions from subsequent runs are concatenated. Thus, the predicted lead year 4-10 time series consists of a concatenation of predictions from the runs initialized in 1960, 1961, 1962, 1963, ..., covering the years 1964-1970, 1965-1971, 1966-1972, 1967-1973, ... .

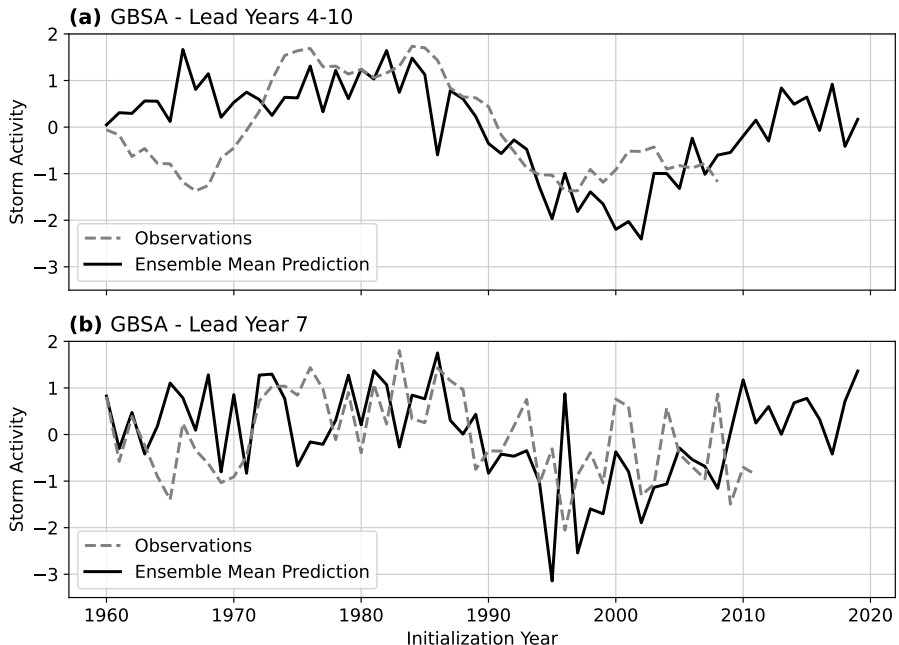

**Figure A1.** Exemplary time series of ensemble mean predictions (black, solid) and corresponding observations (grey, dashed) of German Bight storm activity (GBSA) for lead years 4-10 **(a)** and lead year 7 **(b)**.

*Data availability.* ERA5 reanalysis products that were used to support this study are available from the Copernicus Data Store under https://cds.climate.copernicus.eu/cdsapp#!/search?type=dataset. Three-hourly German Bight MSLP output data from the decadal prediction system are available under https://cera-www.dkrz.de/WDCC/ui/cerasearch/entry?acronym=DKRZ_LTA_1075_ds00011. Seasonal means of North Atlantic MSLP from the decadal prediction system are available under https://cera-www.dkrz.de/WDCC/ui/cerasearch/entry?acronym=DKRZ_LTA_1075_ds00012. Computed German Bight storm activity time series are available under https://cera-www.dkrz.de/WDCC/ui/cerasearch/entry?acronym=DKRZ_LTA_1075_ds00013. Selected global three-hourly atmospheric output from the decadal prediction system is available under https://cera-www.dkrz.de/WDCC/ui/cerasearch/entry?acronym=DKRZ_LTA_1075_ds00014.

*Author contributions.* DK, RW and JB conceived and designed the study. SB carried out the MPI-ESM hindcast experiments and contributed model data. DK, SB, PP, RW and JB analyzed and discussed the results. DK created the figures and wrote the manuscript with contribution from all co-authors.

*Competing interests.* The authors declare that they have no conflict of interest.

*Acknowledgements.* This work has been developed in the project WAKOS – Wasser an den Küsten Ostfrieslands. WAKOS is financed with funding provided by the German Federal Ministry of Education and Research (BMBF; Förderkennzeichen 01LR2003A). JB and PP were funded by the Deutsche Forschungsgemeinschaft (DFG, German Research Foundation) under Germany's Excellence Strategy—EXC 2037 'CLICCS - Climate, Climatic Change, and Society'—Project Number: 390683824, contribution to the Center for Earth System Research and Sustainability (CEN) of Universität Hamburg. JB and SB were supported by Copernicus Climate Change Service, funded by the EU, under contracts C3S-330, C3S2-370. We thank the German Computing Center (DKRZ) for providing their computing resources.

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
