# Peer review of "Skillful Decadal Prediction of German Bight Storm Activity"

_EGUsphere, 2022_

## Referee Comment (RC2)

**Report on *Skillful Decadal Prediction of German Bight Storm Activity**

a manuscript submitted to NHESS by
Daniel Krieger, Sebastian Brune, Patrick Pieper, Ralf Weisse1, and Johanna Baehr

May 26, 2022

**1  General Remarks**

In their study *Skillful Decadal Prediction of German Bight Storm Activity*, the authors aim at adressing a relevant and interesting aspect: a near term climate prediction of storm activity in the German Bight. Language and structure of the manuscript is mostly ok and it fits in the scope of the Journal. However, the description of data and methods is not sufficiently clear and the language is not sufficiently precise. Some concepts and steps in the analysis are described in a somewhat sloppy way and I expect that it will be difficult to reproduce the work described here. Furthermore, the way some methods are used does not seem to be adequate in some cases. The basis for drawing conclusions is thus not robust and the manuscript needs major improvements in several respects before it can be published. I suggest that the authors have a look on the comments and suggestions below. When ever possible, I suggested already solutions to the comments and questions I have in oder to make improving the work as easy as possible. However, there is still a lot of work involved.

**2  Major comments**

**2.1  Conclusions**

I list some of the conclusions drawn in the manuscript and comment on them.

- "Over the North Atlantic region certain regions with significant skill emerge, but the skill is dependent on the length of the averaging window." and some sentences later: "We hypothesize that this lead time dependency might be attributable to the filtering of high-frequency variability by the lnger averaging window, in combination with the model's ability to better predict the inderlying low-frequency oscillations in the large-scale circulation." There is effect from estimating correlation coefficients from autocorrelated series which might – at least partially – account for this effect, see Sec. 2.4.1. The authors should take this into account for their conclusion.

- "... the DPS generates skillful probabilistic predictions for extreme low and high ..." and "As this stands in contrast to the deterministic predictability of winter MSLP anomalies, we want to emphasize that we do not have a convincing explanation for this ..." I suggest to discuss this in relation to the effect of the choice of the reference forecasts as described in Sec. 2.4.3 and the above mentioned effecte of estimating correlations from smoothed series.

- "Highly aggregated probabilistic skill scores[1], which aim at incorporating the model performance for various categories into one single value, might underestimate the capabilities to predict extrmes ..." I suggest to consider the discussion in Sec. 2.4.2 and revisit this conclusion. .

I am surprised that you have not discussed the influence of a drift or initialization shock on your forecasts.

**2.2  Terminology**

Throughout the manuscript the authors use the term "skill" in its colloquiual meaning as "ability to do something" and also in its special meaning within the frame of forecast verification as the value of a skill core. Also other members of the community use this unequivocal way of using the word "skill". However, I think it does lead to confusion. For example, it leads the authors to comparing "skill" of a deterministic forecast (measured with anomaly correlation, which is not a skill score but only a part of an accuracy measure) to "skill" of a probabilistic forecast (measured with the Brier skill score). This is not meaningful, see also below.

You frequently use "deterministic skill" and "probabilistic skill". However, not the skill is probabilistic or not; it is the forecast which is probabilistic or not.

**2.3  Structure**

The section 2.2.2 "Pressure reduction and geostrophic wind" should be renamed to "Geostrophic Wind and German Bight Storm Aktivity" or only "German Bight Storm Aktivity" as this is the goal, as far as I understand. Deriving MSLP and the geostrophic wind are only means to arrive at the GBSA, right? I suggest to show also a time series and a histogram of GBSA.

Maybe you should subdivide Sec. 2.3 into 2.3.1 "Anomaly correlation" and 2.3.2 "Brier Skill Score" to emphasize that these are two really different concepts.

The text from line 304 onwards might also fit well in to the discussion section.

**2.4  Statistical concepts**

**2.4.1  Anomaly correlation**

When introducing the correlation coefficient, you do also mention the fundamentals of your significance test, which I consider as important! However, I have some doubts that
* * *
[1]I assume that the RPS is meant.

this concept will work here as described. Is there an assumption on the distribution of the $f_i$ and $o_i$ for deriving confidence intervals via the $z$-transform? If so, is this assumption reasonably well fulfilled here?

Is the significant test a two-sided or a one-sided test? I assume a two-sided so as you mark also negative ACC as significant but it would be good to clearify.

Later, you use this correlation coefficient for measuring the association of two time series obtained from averaged (over a period of years) forecast and observations. This implies, that successive values $o_i$ (and $f_i$) which enters the estimation of your correlation coefficient are not independent. This reduces the effective number of data points (degrees of freedom) and thus, you have less than $N$ independent data points. This alters your confidence intervals (or critical values for your significance test). In Fig. 1 I give a simple example with associated code in Appendix A. You can see two effects

[Figure]

Figure 1: Example for changing distribution of the correlation coefficient with increasing size of an averaging window. Correlation coefficients are obtained for 100 series of length 1000 data points. Left with no association between the two series, right with a linear association between the series. See Appendix A for the code.

here: i) with increasing size of the averaging window (increasing autocorrelation) the distribution of the correlation coeffcients gets broader (see Fig. 1, left) as there is less information introduced with every new paar of data as they become increasingly dependent with increasing averaging window. This effecte should be reflected in calculating

the significance of your correlation (see **?**). ii) in case there is some association between your forecast and observation, this association will seemingly become stronger for increasing averaging periods (see Fig. 1, right). This might be (at least partially) the reason for your observation that longer averaging periods lead to increased correlation (your Fig. 2).

**2.4.2 Nature of the probabilistic forecast and Brier score**

It is necessary to motivate the discrepancy of obtaining probabilistic forecasts for *three* categories and the use of the Brier score (evaluating *two* categories). If your aim is to issue probabilistic forecasts for the three categories you define, than this forecast can NOT be evaluated with the simple Brier score. Instead, you need the extention of the Brier score to three categories (RPS). If your aim is to issue a dichotomous forecast for the probability of being e.g. in the upper category, than the Brier score is adequate. However, then there is no need to define three categories. Currently one could have the impression that you want to issue a three-category probabilistic forecast but choose to evaluate the *simpler* dichotomous forecast (three times). Juggling three times with two balls is simpler than juggling with three balls. Simpler tasks are likely to lead to better scores. Please consider this when discussing the "neccessity to evaluate the skill for each prediction category individually" in line 310 ff.

Observations, derived values and forecasts seem to be "standardized". That would imply that using $\pm 1$ leads to consistent three categories over all the time series used. It would have been inconsistent If you instead had choosen fixed values $\mu \pm \sigma$ from the observations and use the same fixed values for the forecast. I think, it is well done here, but I suggest to clarify this explicitly. You mention in l.188 that "all" series are standardized. To me, that could refer also only to all observation and derived series.

Please also mention how the probabilistic forecasts for the categories are obtained. I guess this is from counting members with forecasts falling in one of the three categories, right? It is also important to report how you obtain the forecasts for averages of certain periods. Are the series obtained from sliding averages for longer periods also "standardized" again? Or are these categorized according to the individual lead years by their original value? This should make a difference!

It seems somewhat arbitrary that you choose the period of lead years 4-10 for your forecast. Please motivate that.

**2.4.3 Reference forecast**

There are a few canonical reference forecasts in meteorology, e.g. the climatological forecast, the persistence forecast and also a random forecast. When defining these for simple one-step-ahead forecasts, we can easily derive particular characteristics for some skill scores using the above mentioned references, see e.g. **?**. Mostly, we have also a good intuition on how to interpret these reference forecasts. The persistence forecasts is motivated by the fact that there is frequently persistence in the weather system and a simple forecast for tomorrows weather is the weather of today. A forecast which cannot

beat this persistence referene forecast is of no value. In the case of weather forecast, we expect that the persistence forecast outperforms the climatological forecast because of the persistence of weather systems. If we use todays weather to issue persistence forecast with increasing lead time, it seems plausible that the this persistence forecast performs worse with increasing lead time because the basic assumption of persistent weather is less likely to hold for increasing lead time. Looking at your Fig. 4, this effect might play a role when you identify increasing BSS (reference persistence forecast) with lead time. You should discuss this effect when revisiting your conclusions.

For a non-probabilistic forecast for a quantity $X$ with probability distribution $F(x)$, a random foecast can be generated by drawing a random number from $F(x)$. This is a different value for each time you issue this forecast. For a dichotomous probabilistic forecast, a random forecast $f \in [0, 1]$ could be a uniformly distributed random number on $[0, 1]$. I would not call forecast with a fixed value at e.g. $f = 0.5$ a random probabilistic forecast. Fixed probability forecasts are common for the climatological forecast. Here your forecast is the climatological occurrence probability $f = \hat{p}$. I find the interpreation of your "random forecast" difficult. In particular when saying (l.298) that "random guessing" is performing better than "persistence". That indicates that there is something wrong with the common interpretation of either the "random" or the "persistence" forecast, or probably with both. I interpret your "random forecast" as a climatological forecast with a non-adequate estimate of the climatological occurrence probability.

To be more consistent, you could use the same reference forecasts for the non-probabilistic and the probabilistic forecasts. I suggest that the climatology would be a good reference. The MSE might also be adequate for a scoring function for your non-probabilistic forecast.

**2.5   Associated to lines in the manuscript**

**l.9**  "skill of the probabilistic predictions for high and low storm activity notably exceeds the deterministic skill" This is a comparison by numbers of two different measures of forecast quality for two forecasts of very different nature (non-probabilistic and probabilistic). Such a comparison is not meaningful. It is like saying 0.6 Kg cheese cake outperforms 0.5l water because 0.6 is larger than 0.5.

**l.13**  "may greatly benefit" can you give an example how planning and management can benefit from predictions on the decadal time scale?

**l.32**  "While Kruschke ..."  As I understand the situation, Kruschke et al. evaluated a 3-category probabilistic forecast using the ranked probability score. Here, you suggest to evaluate a dichotomous (2-category) probabilistic forecast. The task you choose to address is a lot simpler than the task addressed in Kruschke et al. It is easier to forecast probabilities for 2 categories as this forecast contains less information. If you repeat this task 3 times, it remains an easier task (see my juggling example above). Both strategies (issuing a 2-category and a 3-category forecast) are valid. Which one to chose depends on the goals of your study. Please

explain why you define 3 categories but chose to evaluate a 2-categories forecast. Please note that even if you devide your forecast into three categories, you evaluate with the Brier score only the ability of the model to forecast the probability for being in one categorie or in one of the other two.

**l.62** "The concept of a probabilisitic approach is the presumption that a shift in the ensemble distribution can be used to predict the likeihoods of actual shifts in climatic variables." Please clarify what you mean with this sentence. From my point of view, the concept of probabilistic predictions is to include the uncertainty in the forecast while this is completely ignored for non-probabilistic prediction.

**l.74** "... skill is quantified by correlating ..." The correlation coefficient measures the linear association between two series. It is part of the MSE (a score masuring accuracy) and also part of the associated MSE skill score (measuring skill with respect to the climatological forecast). Only in special cases the MSE skill score with the climatological forecast as a reference is proportional to the correlation coefficient between forecast and observation, see **?** or **?**.

**l.81** Explain what you mean with "standardized 95th percentile". Later in the manuscript you describe "standardization" as the $z$-transformation $z = (x - \bar{x})/sd(x)$. I assume that this is what you mean by "standardization". Please describe here what you do, why you do it and why this is adequate. It might help to show a histogram of wind speeds and/or the 0.95 quantile of wind speeds.

**l.146** Here, I suggest to that you describe in some detail HOW you generate the time series. Do you use the the gradient from a plane through 3 grid points as in Krueger et al? Or is there some averaging involved as you mentioned in l.54?

**l.147** Explain the standardized annual 0.95 quantile and show a time series and histogram plot.

**l.154** Maybe you want to change the title to "Evaluation of forecast/model performance"

**l.155** In fact, you obtain a skill measure (Brier skill score) and a measure of linear association, which I would not call "skill", see Sec. 2.2.

**l.166** Please include in this sentence that you can evaluate *only* dichotomous probabilisitc predictions (i.e. probabilistic predictions for two categories) with the Brier score. The natural extention of the Brier score for more categories is the RPS and for continuous forecasts the CRPS.

**l.173** briefly explain the bootstrap approach used here. What is sampled? With replacement?

**[l.174** ] "... test whether skill scores are significantly different from the reference." $\rightarrow$ either say "test whether skill scores are significantly different from zero." or "test whether model performance is significantly different from the reference".

**l.176** "The individual Brier Scores BS are defined..." → "The Brier Score BS is defined as ..." (In fact, the equation you give is already the average Brier Score over $N$ forecasts.

**l.179** ".. the predicted event .." happened. → "... the event happened". The observation can be 1 even if you have not predicted the event.

**l.180** This is a probabilistic prediction, it can be better or worse but not "correct" or "wrong". Change the sentence accordingly using the range of the BS and its negative orientation.

**l.182** The "random guessing" does not lead to $f = 0.5$, see Sec. 2.4.3.

**l.185** if all your series are "standardized" the $\mu \pm \sigma$ interval should be $\pm 1$. If not, you need to give the $\mu$ as well.

**l.189** GBSA is derived from spatially averaged MSLP gradients? From your description, I thought it is derived from a gradient of the plane through three grid points. Clarification needed.

**l.188** is GBSA also standardized? Clarify.

**l.194** "... persistence prediction of storm activity is generated by taking an average storm activity .." but this must be a probabilistic forecast between 0 and 1, not a storm activity. Clarify.

**l.192** ".. we use a persistence prediction.." it is rather a "kind of persistence". Murphy used it in a clearly defined way. In your case, the quality of the "persistence" reference depends on lead time, see Sec.2.4.3. You might want to check the BS of the persistence as a function of lead time (plot).

**l.204** please be more precise. Positiv anomaly correlation is not equal to a positive skill score.

**l.205** Here you relate negative (and significant) correlation (from your Fig. 1(a)??) to "significant skill". Please clarify.

**l.209** same here.

**l.210** "... overall magnitude of the ACC is lower .." please be more precise. Do you mean that absolute values are on average lager?

**l.211** is that statement not a repition from l.208?

**l.212** please relate this finding to my remark in Sec. 2.4.1.

**l.262** please discuss the performance of the reference prediction (persistence), ideally with a plot depending on lead time.

**l.263** please reconsider the comparison with the "random prediction" under consideration of my remark in Sec. 2.4.3.

**l.265** please define what you mean with "absolute skill"

**l.266** reconsider the comparison to "random guessing"

**l.267** it is correct (and trivial) that the "extreme anomaly events" (in your case those above $\mu + 1\sigma$) occur on average at a rate of $(1 - 0.68)/2 = 0.16 < 0.5$.

**l.297** I agree with the low performance of the reference and suggest that you use climatology as a reference. That would challenge the model more than your "random guessing"

**l.311** please reconsider this paragraph with respect to the discussion in Sec. 2.4.2.

**l.316** maybe a plot of the BS for the reference helps to understand that.

**l.323** "random guessing is ill-suited" and "therefore persistence ranges among the most appropriate references", see Sec. 2.4.3. I suggest climatology.

**l.325** "Our DPS is particularly valuable at lead times during which persistence forecasts are sufficiently poor". That does relate to the discussion of the interpretability of the persistence forecast in Sec. 2.4.3. I would rather discuss this as a deficit in the choice of the reference forecast and *not* as a "valuable" aspect of the DPS.

**l.335** I am not sure how relevant the assumption of normal distributed quantities is. Certainly, you only get a standard normal distribution if your variables are sufficiently normal before standardization. You can get rid of this assumption if you use the 0.16 and the 0.84 quantile of the distribution to categorize your events instead of $\mu \pm \sigma$.

**3   Minor comments**

**l.9** "short lead years" $\rightarrow$ "short aggregation periods"

**l.10** "deterministic skill" not the "skill" (or performance) is deterministic, but the forecast. $\rightarrow$ "performance of the deterministic forecast". But I recomment to *not* compare these two anyway, see above.

**l.11** "... skillfull decadal predictions of regional storm activity can be viable .." $\rightarrow$ "decadal predictions of ... can be skillfull"

**l.16** "... attributed to the anthropogenically caused global warming trend ..." $\rightarrow$ "... attributed to the anthropogenic global warming ..." (Okkhams Razor).

**l.23** "... great potential value in moving ..." $\rightarrow$ "... great potential in moving ..."

**l.23** ".. for each separate category." → "for each category"

**l.53** "..., since averaging over a small area preserves much of the spatial variability .." please specify what you are doing here. Spatial averaging does not preserve spatial variability. The opposite is the case.

**l.56** "... besides the choice of parameters ..." → "besides the choice of variables"

**l.62** "... benefits the generation of probabilistic predictions." → "is favourable for probabilistic predictions."

**l.64** "... of the prediction distribution ..." → "predictive distribution"

**l.65** "viable" → "feasible"

**l.73** and in other places "... deterministic/probabilistic skill ..." → "skill/performance of a deterministic/probabilistic forecast"

**l.86** ".. derived observational time series" What do you mean by that? If you derive a quantity from other quantities, this cannot be an observed series. Do you mean, that the series under question been derived from observations?

**l.109** "constrict" → "constrain"

**l.251** "lead year ranges" → "averaging periods"

**l.286** what is the "ensemble BSS"

**l.290** What is a "skill difference". Skill is already the difference of an accuracy score of a model and a reference (scaled by the difference of the perfect forecast and the reference).

**l.291** "most valuable at skillfully predicting" → "valuable at predicting"

**l.292** "which demonstrated significantly skill" → "which significantly demonstrated positiv skill"

**l.302** ".. not viable to skillfully predict .." → "... is not skillful in predicting ..."

**l.319** "overwhelmingly significant" does not exist. A result is either significant on a certain level or it is not. Less or more significant does not exist. But you can give significance on various levels.

**l.351** "lead range length" → "averaging period"

**l.352** "non-single lead times" → "averaging periods larger 1 year"

**A    Simulation experiment for correlation of smoothed time series**

The following code gives a simple example written in R of how the distribution of the correlation coefficient might change if the two series are smoothed with a running window. In the example, I use a simple autoregressive process to mimic the observations and the forecast.

```r
library(zoo)
N <- 1000
M <- 100
a <- 0.3
mas <- c(1,2,3,4,5,6,7,8,9)
cors  <- cors.2 <- matrix(NA,ncol=length(as),nrow=M)
names(cors) <- as

for(i in 1:length(mas)){
    for(j in 1:M){
        o <- arima.sim(list(ar=a),N)
        f <- arima.sim(list(ar=a),N)
        f.2 <- o+rnorm(N,sd=3)
        cors[j,i] <- cor(rollapply(o,mas[i],mean),rollapply(f,mas[i],mean))
        cors.2[j,i] <- cor(rollapply(o,mas[i],mean),rollapply(f.2,mas[i],mean))
    }
}
layout(matrix(1:2,ncol=2))
boxplot(cors,xlab="size of averaging window",ylab="correlation coefficient")
boxplot(cors.2,xlab="size of averaging window",ylab="correlation coefficient")
layout(1)
```

---

## Author Comment (AC1)

**Response to Reviewer #1**

We sincerely thank Reviewer #1 for their constructive and insightful comments on our manuscript *Skillful Decadal Prediction of German Bight Storm Activity*. The comments greatly helped us to improve the manuscript and clarify key points.
In the following, we will give a point-by-point response to the reviewer's comments and describe how we plan to address the issues raised.

**Data**

**D1** Just to clarify, you are not using the MiKlip data, but have constructed your own decadal prediction system? I was not sure until I got to line 102...

> Our system is indeed based on one developed within MiKlip, namely the "EnKF" system as described in Polkova et al. 2019. However, this system should not be confused with one of the central prediction systems used during the actual life time of MiKlip. These systems all used oceanic and atmospheric nudging for assimilation and lagged initialization for the ensemble generation.
In contrast to the "EnKF" system within MiKlip, our prediction system includes CMIP6 instead of CMIP5 external forcing, and the hindcasts are run with a total of 80 members, members 17-80 also with 3-hourly output. These 64 members are analyzed in our study.
We will add two sentences at the beginning of the corresponding paragraph.

**D2** I do not quite understand how you constructed the 64-member ensemble (L104-111). Please describe this in more detail.

> The initialization of five members each are derived from one assimilation member, the only difference between those five members coming from the perturbation applied to the horizontal diffusion coefficient in the stratosphere. With a 16-member assimilation, this results in 5x16=80 members. However, 3-hourly output is only available for members 17 to 80, which comprise the 64-member ensemble used in our study.
We will adapt the description within the paragraph with a more distinct explanation.

**D3** Please clarify which decadal runs you chose. If you are looking at the period 1961-2018, did you select all runs that include those years regardless of the lead time, or is the last run you selected the one that was initialized in 2008?

> We always select the maximum number of possible runs for each lead year range. This means that, for example, the last run used for a lead year 1 evaluation is the one initialized in November 2017, whereas for a lead year 10 evaluation the last run would be the one initialized in November 2008. For longer averaging periods, the last lead year is decisive, so the lead year 4-10 evaluation considers all runs up to 2008, whereas the lead year 4-6 evaluation includes runs up to 2012.
We will add two sentences at the end of the corresponding paragraph to clarify which runs we use.

**Methods**

**M1** *Lead times, part 1:* The selection of lead times seems somewhat arbitrary. Why did you choose 4-10 and 7 and not 1-7 and 4 or 2-8 and 5 ...? Have you checked whether your results/conclusions would be different with a different choice of lead time?

> We thank the reviewer for raising this issue. We are aware that choosing lead years 4-10 and 7 is quite arbitrary, and it would be equally valid to choose 2-8 and 5, or 1-7 and 4. We also checked other combinations of the same averaging period and found that similar general conclusions can be drawn from these lead times (see Fig. 1). We will clarify that for reasons of brevity we just show one example for short (7) and long (4-10) averaging periods, respectively, but the conclusions hold for other lead time combinations as well.

[Figure]

**Fig. 1:** Gridpoint-wise anomaly correlation coefficient (ACC) for DJF MSLP anomalies between the hindcast ensemble mean and ERA5 for lead years 1-7 **(a)** and lead year 4 **(b)**. Stippling indicates significant correlations (p<=0.05), determined through a 1000-fold bootstrapping with replacement.

**M2** *Lead times, part 2:* In L126ff, you state that you focus on lead years 4-10 and 7. However, this only applies to the MSLP anomalies, since you show all possible lead year ranges for GBSA. Please be more specific in this regard.

> We will clarify that we only show lead years 4-10 and 7 for MSLP, but all combinations for GBSA.

**M3** *Pressure reduction:* Is this a standard procedure for calculating MSLP from modelled surface pressure? Could you add a reference for equation 1? Does it affect the comparability of your results if you use direct MSLP for one half of the ensemble and calculate MSLP for the other half?

> We will add a reference for the pressure reduction formula, which is based on the US standard atmosphere and a fixed air density, as described in Alexandersson et al. (1998) and Krueger et al. (2019). We also performed a consistency check (Fig. 2) to quantify the MSLP difference between direct and derived output and found that it is negligible for low elevations.

[Figure]

**Fig. 2:** Difference between manually reduced MSLP and model-output MSLP for one exemplary ensemble member, shown as a 10 year mean (2021-2030) of data from the 2020 initialization. Red colors indicate regions where the manual reduction results in higher MSLP than the automated model output.

**M4** *Region of interest:* Please clarify that you are analysing MSLP anomalies for the entire North Atlantic basin (including the German Bight), whereas the GBSA analyses focus only on the German Bight.

> We thank the reviewer for making us aware that this is unclear. We will clarify this paragraph and explicitly state that we investigate MSLP for the whole North Atlantic basin.

**M5** *Selection of grid points (L140-144):* This information refers to the generation of GBSA time series, correct? If so, either integrate it in the respective paragraph (L146ff) or clarify why you need to select three grid points. At the moment, the whole paragraph comes a bit out of nowhere, without a clear link to the preceding/subsequent paragraphs...

> We thank the reviewer for this suggestion. We agree that the structure of this part of the method section needs improvement to make it more comprehensible. We will rewrite and restructure the respective paragraphs.

**M6** *Generation of GBSA time series:* Did I understand correctly that the time series cover the whole period 1960-2018, while you only use the period 1961-2010 for the standardization?

> That is correct. We base the choice of 1961-2010 as a reference period on Krieger et al. (2020), who also used 1961-2010 to standardize the timeseries. We decided to adapt this reference period in order to introduce as few inhomogeneities as possible.

**M7** *Prediction skill:* Please add a short explanation of why it is important to consider both deterministic and probabilistic skill scores when assessing the skill of a decadal prediction system.

> We thank the reviewer for this suggestion. We will add a paragraph on the benefits of assessing the forecast skill of two different prediction methods.

**M8** *ACC:* Although this should be common knowledge, please add the possible range of ACC and an explanation of what the different values mean.

> We will add a sentence on the characteristics of the ACC and the possible range in the respective paragraph.

**M9** *ACC versus BS:* Be careful when using f and o in equations 2 and 4. You chose the same letters, but they have different meanings (value for ACC, probability for BS). Consider replacing f and o in equation 4 with capital letters.

> We thank the reviewer for making us aware of this unclear nomenclature. We will change the variables in Equation 4 to capital letters to avoid further confusion.

**M10** *Choice of BSS:* Out of curiosity – why did you choose the BSS rather than the ranked probability skill score (RPSS)? Since you are interested in three categories (low/normal/high), the RPSS seems the more natural choice to me as it also contains some information about the distance between model and observations.

> We chose the BSS instead of the RPSS as we wanted to investigate whether the model is particularly skillful in predicting one of the three defined categories. By calculating three distinct skill scores for the dichotomous forecasts *high/not high, low/not low, moderate/not moderate*, we want to demonstrate the added value for forecasts of extreme periods, and the inability of the model for forecasts of moderate activity periods. This distinction would not have been possible by calculating the RPSS, which incorporates the skill for every distinct category into one single measure.
We will clarify our intention to use the BSS instead of the RPSS in the respective paragraphs.

**Results**

**R1** *Some thoughts on L234-242:* Could it be that the initialisation has a "negative" impact in the first years (some kind of initialisation shock) – which would explain why the predictive skill is highest for lead time ranges starting in year 3 and 4? This would also fit (to some extent) to previous studies on wind-related variables like Kruschke et al. (2014) or Moemken et al. (2016). However, these studies use uninitialized historical simulations as reference and not persistence...
For temperature, several studies show high predictive skill for later/longer lead times (e.g. Feldmann et al., 2019). This increase seems to originate mainly from the longterm climate trend. However, I have never heard of the importance of climate trend for decadal predictions of wind-based parameters...

> We thank the reviewer for their thoughts on a possible initialization shock. In fact, the predicted geostrophic winds are lowest for lead years 3-5, and highest for lead year 1 (Fig. 3). While we use lead year 1 means and standard deviations to derive standardized GBSA from the absolute geostrophic winds for all data, we also tested whether standardizing each lead year with its respective mean and standard deviation has a notable effect on the results. We find that the ACC between model and observation is almost unaffected by the choice of our standardization reference (Fig. 4).

Nevertheless, we will expand the discussion of our results with a paragraph on the effects of a possible initialization shock.

Regarding the climate trend, we agree that the prediction skill for longer lead times can be greatly impacted by the presence of a trend. However, as the reviewer already correctly states, there is little agreement on the response of storm activity to future climate change. Additionally, observational records indicate that, so far, there has not been a significant climate signal in storm activity in our study region, which leads us to believe that long-term trends don't play a major role in the prediction skill here.

[Figure]

**Fig. 3:** Absolute 95th annual percentiles of predicted geostrophic winds per lead year. Red dots represent individual members and initialization years, black dots show the ensemble mean for each lead year.

[Figure]

**Fig. 4:** ACC between observations and ensemble mean predictions of German Bight storm activity (GBSA) for all combinations of start and end lead years. **Left:** GBSA predictions are based on lead years that are individually standardized by their respective means and standard deviations, i.e., absolute geostrophic winds for lead year 5 are standardized by subtracting the mean and dividing by the standard deviation of lead year 5. **Right:** GBSA predictions are always based on a standardization with respect to lead year 1, i.e., absolute geostrophic winds for lead year 5 are standardized by subtracting the mean and dividing by the standard deviation of lead year 1, like in the original manuscript.

**R2** *L304-338:* These paragraphs seem to be more of a general discussion of your results and are not really related to the rest of section 3.2.2. Therefore, it might make sense to introduce a new section (3.3 Discussion) or new chapter (4. Discussion) for this part of the manuscript.

> We will separate the paragraphs discussing our results from the result description and create a separate section for discussion only.

**R3** *Persistence as reference:* Many studies dealing with decadal prediction systems use uninitialized historical simulations of the same model or simple climatology as reference. Is there any particular reason why you have not tried this as well? Please do not get me wrong – I think it is a strength of your study that you consider persistence and random guessing. It just makes it harder to compare your results with other studies on decadal predictions.

> After revisiting the manuscript and reviews, we also see the need to discuss the performance of the model against climatology, as climatology proves to be a tougher challenge than random guessing. We originally opted for persistence and random guessing to not overload the manuscript with a large number of different comparisons, but we agree that using climatology as an additional reference simplifies the comparison of our results with those from other studies. We will restructure the results section and add comparisons to climatology where we see fit.

**Figures**

**F1** For readers unfamiliar with Germany (and the German Bight in particular), it might be helpful to include a figure showing the region of interest. In this, you could also mark the grid points given in Table 1.

> We agree that a map will be helpful and will add one to the methods section.

**F2** Figure 2: Please add some explanation in the text (L226-230) about the structure of the plot (that it shows all possible lead time combinations etc.).

> We will add a short introduction on the structure of the matrix plots before summarizing the key findings of Figure 2.

**F3** Consider simplifying the captions of Figures 5 and 6 (the same applies to B3 and B4) by saying something like "Same as Figure 4, but for …".

> We will shorten the repetitive figure captions wherever applicable.

**Specific comments**

> We will address all minor comments noted by the reviewer as suggested.

**References**

Alexandersson, H. et al. (1998): Long-term variations of the storm climate over NW Europe, The Global Atmosphere and Ocean System, 6.

Krueger, O. et al. (2019): Northeast Atlantic Storm Activity and Its Uncertainty from the Late Nineteenth to the Twenty-First Century, Journal of Climate, 32, 1919–1931, doi:10.1175/JCLI-D-18-0505.1.

Polkova, I. et al. (2019): Initialization and ensemble generation for decadal climate predictions: A comparison of different methods. Journal of Advances in Modeling Earth Systems 11 (1), 149-172, doi:10.1029/2018MS001439.

---

## Author Comment (AC2)

**Response to Reviewer #2**

We sincerely thank Reviewer #2 for their constructive and insightful comments on our manuscript *Skillful Decadal Prediction of German Bight Storm Activity*. The comments greatly helped us to improve the manuscript and clarify key points.
In the following, we will give a point-by-point response to the reviewer's comments and describe how we plan to address the issues raised.

**2.1 Conclusions**

**2.1A** On the effect of autocorrelated time series on increased correlation coefficients for longer averaging periods

> We agree with the reviewer. We will expand our discussion and conclusion sections with additional thoughts on the effect of averaging window length on the correlation of associated time series. Please also see our reply to comment 2.4.1B.

**2.1B** On the choice of reference forecasts and the effect of estimating correlations from smoothed timeseries

> We agree that there is a need to discuss the effect of the choice of reference in greater details. We will enhance the conclusions to also include the lead-time dependence of persistence (comment 2.4.3A) and a comparison with climatology (comment 2.4.3C).

**2.1C** Revisiting the conclusion that is based on the choice of the Brier Skill Score instead of the RPSS

> The reviewer is correct in assuming that we refer to the RPS/RPSS when mentioning "highly aggregated probabilistic skill scores". We will enhance this section of the conclusions to bring in our intent and elaborate more on the differences between the general concepts of the RPS and the BS, which we also explain in our reply to comment 2.4.2A.

**2.1D** On a possible initialization shock or model drift

> We thank the reviewer for suggesting the possibility of an initialization shock or model drift. In fact, the predicted geostrophic winds are lowest for lead years 3-5, and highest for lead year 1 (Fig. 1). While we use lead year 1 means and standard deviations to derive standardized GBSA from the absolute geostrophic winds for all data, we also tested whether standardizing each lead year with its respective mean and standard deviation has a notable effect on the results. We find that the ACC between model and observation is almost unaffected by the choice of our standardization reference (Fig. 2).
Nevertheless, we will expand the discussion of our results with a paragraph on the effects of a possible initialization shock.

[Figure]

**Fig. 1:** Absolute 95th annual percentiles of predicted geostrophic winds per lead year. Red dots represent individual members and initialization years, black dots show the ensemble mean for each lead year.

[Figure]

**Fig. 2:** ACC between observations and ensemble mean predictions of German Bight storm activity (GBSA) for all combinations of start and end lead years. **Left:** GBSA predictions are based on lead years that are individually standardized by their respective means and standard deviations, i.e., absolute geostrophic winds for lead year 5 are standardized by subtracting the mean and dividing by the standard deviation of lead year 5. **Right:** GBSA predictions are always based on a standardization with respect to lead year 1, i.e., absolute geostrophic winds for lead year 5 are standardized by subtracting the mean and dividing by the standard deviation of lead year 1, like in the original manuscript.

**2.2 Terminology**

**2.2A** On the usage of the term "skill"

> The reviewer is correct that the correlation coefficient per se is not a measure of forecast skill, but much rather a measure of linear association. We will refrain from calling the ACC a skill score.

**2.2B** On the usage of the terms "deterministic skill" and "probabilistic skill"

> We agree that the terms "deterministic skill" and "probabilistic skill" are not precise, as "deterministic" and "probabilistic" refer to the forecast types. We will resort to a more precise terminology.

**2.3 Structure**

**2.3A** On the structure of the section on pressure reduction and the derivation of GBSA

> We agree that the title of this subsection needs to be changed to more accurately reflect its scope. We will reword the title and also restructure this subsection.

**2.3B** On subdividing the model evaluation section into two parts.

> We agree that dividing this section makes it clearer to the reader that we are introducing two different concepts here. We will split up the section.

**2.3C** On establishing a separate discussion section.

> We agree that the text from 304 onwards discussed the results rather than describing them. We therefore see the need to create a separate discussion section and will add one in the updated manuscript.

**2.4 Statistical concepts**

**2.4.1 Anomaly correlation**

**2.4.1A** On the effect of autocorrelation on significance

> We agree on this point. Calculating confidence intervals and significance levels via the Fisher-z transformation requires independent samples, an assumption that is not satisfied in our case due to autocorrelation. We will recalculate the significance with a block-bootstrapping approach, as especially the smoothed (multi-year average) time series are heavily autocorrelated.

**2.4.1B** On the association between forecast and observations and its effect on correlation

> We thank the reviewer for the explanation and sample code on the effect of autocorrelation on the correlation coefficient of smoothed time series. While it doesn't explain the entirety of the ACC increase for longer averaging periods, it might account for a part of it. We will add this effect to our discussion and conclusion sections.

**2.4.2 Nature of the probabilistic forecast and Brier score**

**2.4.2A** On the choice of the Brier score instead of the RPS/RPSS

> We thank the reviewer for bringing up the issue of evaluating a 3-category forecast with the Brier (skill) score. We completely agree that the RPS/RPSS is the correct evaluation metric for a 3-category forecast. However, we are not aiming at correctly predicting which category out of the three will occur, but much rather whether the model shows skill for a 2-category forecast with different event thresholds. To use the reviewer's analogy, we are interested how often (out of the three options) the model succeeds in juggling with two balls, not whether the model succeeds in juggling with three balls. While the RPS/RPSS acts as a metric for how well the model performs for a 3-category forecast, it is unable to show whether, for example, a *high vs. no high activity* prediction is more skillful than a *low vs. no low activity* prediction. We totally agree that we need to clarify this intent in order to avoid the impression that we aim at generating a 3-category forecast and appreciate your insightful thoughts on this matter.

**2.4.2B** On the explicit clarification of the "standardization"

> We agree with the reviewer that we need to clarify our standardization process more explicitly. We will rephrase the part of the methods section which introduces the standardization to make this matter clearer.

**2.4.2C** On obtaining the probabilistic forecasts and re-standardization

> We apologize for being vague here. The probabilities are obtained by counting the number of members above/below a category threshold and dividing this number by the total number of members in the ensemble (64). The time series of moving averages for longer periods are standardized again, so that we always compare predicted and observed time series, which by definition have a mean of 0 and a standard deviation of 1. We agree that we need to describe this more thoroughly, as it indeed makes a difference. We will improve the section on standardization to reflect this procedure.

**2.4.2D** On the choice of lead years 4-10

> We agree that the choice of lead years 4-10 (and 7) appears quite arbitrary. We could have also done the analysis for lead years 1-7 and 4 (Fig. 3) and drawn equally valid conclusions from those lead times. We will clarify our intent in the methods and results sections and try to highlight that we only give two examples for reasons of brevity, but draw conclusions for more lead year ranges than just the two shown.

[Figure]

**Fig. 3:** Gridpoint-wise anomaly correlation coefficient (ACC) for DJF MSLP anomalies between the hindcast ensemble mean and ERA5 for lead years 1-7 **(a)** and lead year 4 **(b)**. Stippling indicates significant correlations (p<=0.05), determined through a 1000-fold bootstrapping with replacement.

**2.4.3 Reference forecast**

**2.4.3A** On the lead-time dependence of the skill of persistence forecasts

> We thank the reviewer for their thoughts on the lead-time dependence of persistence. We calculated the Brier Score for persistence (Fig. 4). While there are certain lengths of averaging periods, for which the BS is lower, there is no general decline in BS with increasing lead time or averaging period. We believe that including plots of the Brier Skill for all forecasts and categories would clutter the manuscript. However, we will improve the conclusions by discussing the performance of persistence in more detail.

[Figure]

**Fig. 4:** Brier Score of persistence for all combinations of start and end lead years for high storm activity **(left)** and low storm activity **(right)**. Reddish colors indicate BS lower (= better) than that of a coin-flip forecast, bluish colors indicate BS higher (= worse) than that of a coin-flip forecast.

**2.4.3B** On random forecasts

> We are sorry for causing confusion by our ambiguous use of the term "random forecast". Our intention was to create a reference forecast that always predicts a fixed probability of 50%, not one that randomly selects one of two outcomes every year. We will rename the previous "random forecast" to avoid confusion, and additionally discuss the performance of the model compared to a true random forecast with fixed climatological probabilities (as e.g. described in Wilks, 2011). Please also see our reply to comment 2.4.3C below.

**2.4.3C** On climatology as a reference forecast

> After revisiting the manuscript and taking into account the points raised by multiple reviewers, we agree that there is the need to compare the model to a climatology-based prediction. We originally opted against that to avoid an excessive number of different references in the results section, but we see that this is a more challenging test for the model than a fixed 50% probability forecast. We will test the model against climatology and add the results to the manuscript.

**2.5 Comments on specific lines in the manuscript**

For reasons of brevity, we will refrain from repeating all comments in this document, and instead just refer to the line numbers in the original manuscript.

**9** We thank the reviewer for making us aware of this issue. Our intention behind this statement was to mention that, for certain lead times, a probabilistic forecast can show significant skill, while a deterministic forecast produces insignificant correlation coefficients, and vice versa. We understand that this distinction needs to be made a lot clearer, and we will revise this paragraph accordingly.

**13** We will add an example for the benefit of decadal predictions and a reference to this paragraph.

**32** We will address the methodical difference between evaluating a 3-category forecast with the RPSS (as in Kruschke et al., 2016) and evaluating three separate 2-category forecasts with the BSS and the implications of doing so in greater detail. Please also see our reply to comment 2.4.2A.

**62** We thank the reviewer for bringing up this additional detail. The uncertainty of a forecast is a great advantage of probabilistic predictions.
With the quoted statement, we intended to explain that periods of high and low storm activity might be detectable through changes in the shape of the ensemble distribution and its tails. If the whole distribution shifts towards one direction, the shift should be notable in both a probabilistic and a deterministic (ensemble mean) prediction. However, if the shape of the tails in particular changes, a probabilistic prediction might hold an advantage over the deterministic prediction. We will amend the paragraph to correctly reflect this explanation.

**74** We thank the reviewer for pointing out the difference of using a skill score for deterministic predictions like the MSE and a measure for linear association like the ACC. We believe that the ACC is widely established as a metric to quantify the ability of a climate model ensemble mean to predict the temporal evolution of a quantity. We will therefore opt to keep the ACC as our metric, and instead stop referring to it as "deterministic skill" (compare comment 2.2A).

**81** That is correct. We standardize time series by applying a z-transformation, i.e., by subtracting the mean and dividing by the standard deviation. We will add a clearer definition of the standardization process to the respective paragraphs.
As for the histogram, we agree that a histogram of (absolute) wind speeds will benefit the methods section. However, we believe that it might be more suited to add it to the paragraph on deriving GBSA from model output, mainly because the observational GBSA is based on an average of 18 different standardized time series of geostrophic winds from overlapping triangles. The absolute 95$^{th}$ percentiles vary depending on the size of the individual triangles, preventing any generalization of the absolute wind speeds. For the model output, it might be more consistent to show a histogram of absolute wind speeds.

**146** We apologize for leaving this unclear. We use the MSLP gradient of a plane through three grid points, as it was done in previous studies (e.g., Alexandersson et al., 1998; Krueger et al., 2019). We will rephrase this sentence to avoid misconceptions.

**147** We will enhance the explanation of the concept of using standardized 95$^{th}$ percentiles and add an exemplary histogram of absolute wind speeds to the respective section.

**154** We thank the reviewer for this suggestion and agree that this title would fit the section better. We will update the title of the section.

**155** Please see our reply to comment 2.2A.

**166** We will amend that sentence to include the dichotomous nature of the Brier score. Please see also our reply to comment 2.4.2A.

**173** We apologize for not going into enough detail here. We bootstrap by sampling different forecast/initialization years with replacement. We do not apply the bootstrap to sample ensemble members. We will clarify our approach in this paragraph.

**174** We thank the reviewer for pointing out this imprecise wording. We will amend the sentence to be more accurate.

**176** We will change the sentence to be more accurate.

**179** The reviewer is correct; our phrasing was ambiguous. We will update this sentence following your suggestion.

**180** We thank the reviewer for this suggestion. We will change the wording and elaborate more on the characteristics of the Brier Score.

**182** Please see our reply to comment 2.4.3B.

**185** That is correct. We will remove the sigma from the category thresholds.

**188** Yes, both GBSA time series from the model and the observations are standardized. We will rephrase this paragraph to clarify.

**189** We apologize for causing confusion here. GBSA is derived from the MSLP gradient of a plane through three grid points, as the reviewer correctly notes. The gradient of the plane can be interpreted as the average horizontal MSLP gradient of the triangle spanned by the three grid points, but there is no averaging of different gradients involved in the derivation of GBSA. We will rephrase this sentence to avoid misconceptions.

**192** Please see our reply to comment 2.4.3A.

**194** We apologize for being unclear here. The persistence forecast is not a probabilistic forecast, but rather a deterministic one. We use the average storm activity of the past $n$ years as a persistence forecast for our target lead years and assign it a Brier Score of either 0 or 1, depending on whether the persistence forecast is on the same side of the threshold as the observation or not. In other words, the persistence always forecasts a probability of either 0% or 100% for an event to happen.
We will rephrase the respective paragraphs to clarify this.

**204** Please see our reply to comments 2.2A and 2.2B.

**205/209** The reviewer is right; a significant negative correlation is just useful for predictions when the physical reason behind it is clear, which is not the case here. We will clarify that, while the correlation itself is significant, this is of little use for a skillful prediction.

**210** Yes. We will rephrase that sentence to clarify that we refer to the absolute correlations being larger on average.

**211** Yes. We thank the reviewer for making us aware of that. We will remove the redundant statement.

**212** We thank the reviewer for this suggestion. We will improve the discussion on the effect of averaging periods on ACC.

**262** We will improve the discussion on the lead-year dependence of persistence. Please also see our reply to comment 2.4.3A.

**263** Please see our reply to comment 2.4.3C.

**265** We apologize for this misleading terminology. We will remove the word "absolute", as it does not fit here.

**266** Please see our reply to comment 2.4.3C.

**267** We agree.

**297** Please see our reply to comment 2.4.3C.

**311** We agree with the reviewer that this part of the discussion needs to be enhanced. Please see our reply to comments 2.4.2A and 32.

**316** We thank the reviewer for this suggestion. Please see our reply to comment 2.4.3A.

**323** Please see our reply to comment 2.4.3C.

**325** We agree that it is certainly also a deficit that comes with using persistence as a reference. We think that an improvement in skill over a reference can be seen in two ways, both as a valuable aspect of the DPS and as a deficit of the reference. We will update the respective paragraph to include both viewpoints.

**335** We thank the reviewer for this thought. Using the 0.16 and 0.84 quantiles would indeed eliminate the need for normal distributed quantities. We will include this in the discussion section.

**Minor comments**

> We will address all minor comments noted by the reviewer as suggested.

**References**

Alexandersson, H. et al. (1998): Long-term variations of the storm climate over NW Europe, The Global Atmosphere and Ocean System, 6.

Krueger, O. et al. (2019): Northeast Atlantic Storm Activity and Its Uncertainty from the Late Nineteenth to the Twenty-First Century, Journal of Climate, 32, 1919–1931, doi:10.1175/JCLI-D-18-0505.1.

Kruschke, T. et al. (2016): Probabilistic evaluation of decadal prediction skill regarding Northern Hemisphere winter storms, Meteorologische Zeitschrift, 25, 721–738, doi:10.1127/metz/2015/0641.

Wilks, D.S. (2011): Statistical Methods in the Atmospheric Sciences. 3rd Edition, Academic Press, Oxford.

---

## Author Comment (AC3)

Response to Reviewer #3

We sincerely thank Reviewer #3 for their constructive and insightful comments on our manuscript *Skillful Decadal Prediction of German Bight Storm Activity*. The comments greatly helped us to improve the manuscript and clarify key points.
In the following, we will give a point-by-point response to the reviewer's comments and describe how we plan to address the issues raised.

**Comments**

**C1** One issue is that the paper focusses on some unusual forecast lead times (4-10 years and 7 years) without properly motivating why they use these. It would not seem the most interesting lead times for a user of a storm activity forecast. There is frequent reference to short and long averaging periods and I was not always sure whether that referred specifically to these two periods or had been generalised somehow. But if it is the latter, it is not defined. The language needs to be cleaned up around this.

> We thank the reviewer for making us aware of this ambiguity in the manuscript. Our intent was to show two exemplary lead year ranges, one for short averaging periods, and one for long averaging periods. The analysis could for example have also been done for lead years 1-7 and 4 (Fig. 1), leading to similar conclusions. We will clarify our intent in the methods and results sections and try to highlight that we only give two examples for reasons of brevity, but draw conclusions for more lead year ranges than just the two shown.

[Figure]

**Fig. 1:** Gridpoint-wise anomaly correlation coefficient (ACC) for DJF MSLP anomalies between the hindcast ensemble mean and ERA5 for lead years 1-7 **(a)** and lead year 4 **(b)**. Stippling indicates significant correlations (p<=0.05), determined through a 1000-fold bootstrapping with replacement.

**C2** Figures which lead to firm conclusions are squirreled away in an appendix. I suggest all important figures need to be in the main paper. See minor comments below.

> We agree that the selection of figures in the main body of the manuscripts needs improvement. We will rearrange the figures, so that all figures which support the conclusions can be found in the results section, not in the appendix.

**C3** Another issue is that there is no deterministic skill for mslp anywhere near the German Bight (Figure 1), so how do you explain that you have skill in predicting storm activity there? This needs to be covered in the discussion.

> We apologize for leaving this vague. One way to explain it could be that German Bight storm activity does not depend on the MSLP itself, but on the annual statistics (95[th] percentiles) of the horizontal MSLP gradients, for which the model shows some skill. This might be due to the model being unable to predict fluctuations around the mean, but being able to predict sufficiently large deviations from the mean. We will add a paragraph on this contradiction to the discussion section.

**C4** Negative skill is presented as useful skill. It is true, you could multiply the forecast by -1 and get a good forecast on average. The problem is that the skill is possibly negative due to a poorly modelled teleconnection and if there is an individual year when that teleconnection is not strong, multiplying by -1 could be the wrong thing to do. Better to assume negative skill is not useful even if it is significant.

> The reviewer is right; a significant negative correlation is just useful for predictions when the physical reason behind it is clear, which is not the case here. We will clarify that, while the correlation itself is significant, this is of little use for a skillful prediction.

**C5** Finally, the text often refers to "tails" of the distribution and "extremes" when in fact the data refers to anomalies exceeding 1 sigma, which is neither in the tail or an extreme. These words need to be removed from the text.

> We agree that the terms "tail" and "extremes" can be misleading when talking about +/- 1-sigma events. We will replace these terms with better-suited vocabulary.

**Minor comments**

For reasons of brevity, we will refrain from repeating all comments in this document, and instead just refer to the line numbers in the original manuscript.

**9** Here, "short lead years" should rather read "short averaging periods". We apologize for this mistake and will rephrase this sentence.

**126** We chose two exemplary lead year periods, one for long averaging periods, and one for short averaging periods. There is probably little interest in forecasts for lead years 4-10 specifically, but our intention was to use two cases to bring our point across. We definitely see the need to motivate this choice better and will enhance the respective paragraphs. Please also see our response to comment C1.

**149** We calculated the means and standard deviations for each member separately to account for possible biases/shifts between individual members, and to force each member to be centered around a storm activity of 0. It would be equally valid to allow biases between members and use the full ensemble mean and standard deviation for the derivation of GBSA. Please see the figure below (Fig. 2), showing the model-observation ACC (deterministic skill) after using the individual means and standard deviations (left), as well as after using the full ensemble mean and standard deviation (right). The differences in ACC are negligible. This is also the case for the BSS of the probabilistic forecast.

[Figure]

**Fig. 2:** ACC between observations and ensemble mean predictions of German Bight storm activity (GBSA) for all combinations of start and end lead years. GBSA predictions are based on individually standardizing members with their respective means and standard deviations **(left)**, and on standardizing with the mean/standard deviation of the full ensemble **(right)**.

**162** The reviewer is correct. The Fisher-z method requires independent samples, which we did not account for. We will recalculate the significance with a bootstrapping approach, as especially the smoothed (multi-year average) time series are heavily autocorrelated.

We also agree that adding time series of GBSA would be helpful. However, we want to note that these time series would also likely be for exemplary lead times only, since including time series for all 55 possible lead time combinations would overload the manuscript. We will restructure the methods sections and then re-evaluate the possibility of including exemplary time series of derived GBSA, for example for lead years 4-10 and 7.

**195** We will rephrase this sentence.

**205/211** Please see our response to comment C4.

**224** We will remove "anyhow" from this sentence.

**241** We thank the reviewer for their thoughts on a possible initialization shock. In fact, the predicted geostrophic winds are lowest for lead years 3-5, and highest for lead year 1 (Fig. 3). While we use lead year 1 means and standard deviations to derive standardized GBSA from the absolute geostrophic winds for all data, we also tested whether standardizing each lead year with its respective mean and standard deviation has a notable effect on the results. We find that the ACC between model and observation is almost unaffected by the choice of our standardization reference (compare Fig. 4 and Fig. 2 left). Nevertheless, we will expand the discussion of our results with a paragraph on the effects of a possible initialization shock.

[Figure]

**Fig. 3:** Absolute 95th annual percentiles of predicted geostrophic winds per lead year. Red dots represent individual members and initialization years, black dots show the ensemble mean for each lead year.

[Figure]

**Fig. 4:** ACC between observations and ensemble mean predictions of German Bight storm activity (GBSA) for all combinations of start and end lead years. GBSA predictions are based on lead years that are individually standardized by their respective means and standard deviations, i.e., absolute geostrophic winds for lead year 5 are standardized by subtracting the mean and dividing by the standard deviation of lead year 5, instead of lead year 1 like in the original manuscript.

**246** The reviewer is correct here. If the entire distribution shifts, the shift would also be present in the ensemble mean. The shape of the distribution and specifically the tails needs to change in order for a probabilistic prediction to gain an advantage over the deterministic prediction. We will amend the paragraph to correctly reflect this explanation.

**251** We apologize for being unclear. We will explicitly mention the lead year ranges and then draw conclusions from that.

**260** We will highlight the German Bight on the skill maps.

**271** We thank the reviewer for bringing up this point. We will rephrase our conclusions to be more precise in stating that we only show two specific lead year ranges for reasons of brevity, but have looked at other lead year ranges as well to draw conclusions that are more general. We will also give a clearer definition on short and long lead year periods in the methods section.

**302** We thank the reviewer for that suggestion. Please see our response to comment C2.

**319** We will rephrase that sentence to make it less informal.

**323** Absolutely. After revisiting the manuscript and considering the points raised by multiple reviewers, we agree that there is the need to compare the model to a climatology-based prediction, which would be a prediction that uses the climatological probabilies of a year falling into a certain category. We originally opted against that to avoid an excessive number of different references in the results section, but we see that this is a more challenging test for the model than a fixed 50% probability forecast. We will test the model against climatology and add the results to the manuscript.

**329-338** We will create a separate discussion section to discuss our results, as well as limitations and caveats of the model. Everything from line 304 onwards will be moved to this separate section.

**345** That is correct. We will update this sentence.

**361** We will expand on this paragraph and give a speculation on the origin of the skill for short averaging periods.

**Fig. 1** We will add a figure to the methods section showing a map of the study region and the locations of the three model gridpoints.

---

## Author Response (AR1)

Dear Joaquim Pinto,

Thank you for the opportunity to submit a revised version of our manuscript #EGUSPHERE-2022-288 titled "Skillful Decadal Prediction of German Bight Storm Activity" to *Natural Hazards and Earth System Sciences*. We, the authors, would like to express our gratitude to the three anonymous referees for providing valuable feedback on our manuscript.

We would like to give a point-by-point response to the reviewers' comments in the next section, followed by a list of additional changes that we felt were necessary to improve the quality of the manuscript.
* * *
**Response to Reviewer #1**

**Data**

**D1** Just to clarify, you are not using the MiKlip data, but have constructed your own decadal prediction system? I was not sure until I got to line 102...

> Our system is indeed based on one developed within MiKlip, namely the "EnKF" system as described in Polkova et al. 2019. However, this system should not be confused with one of the central prediction systems used during the actual lifetime of MiKlip. These systems all used oceanic and atmospheric nudging for assimilation and lagged initialization for the ensemble generation.
In contrast to the "EnKF" system within MiKlip, our prediction system includes CMIP6 instead of CMIP5 external forcing, and the hindcasts are run with a total of 80 members, members 17-80 also with 3-hourly output. These 64 members are analyzed in our study.
We expanded the section on our decadal hindcasts to clarify how our prediction system differs from the MiKlip data.

**D2** I do not quite understand how you constructed the 64-member ensemble (L104-111). Please describe this in more detail.

> The initialization of five members each are derived from one assimilation member, the only difference between those five members coming from the perturbation applied to the horizontal diffusion coefficient in the stratosphere. With a 16-member assimilation, this results in 5x16=80 members. However, 3-hourly output is only available for members 17 to 80, which comprise the 64-member ensemble used in our study.
We replaced the description within the paragraph with a more distinct explanation.

**D3** Please clarify which decadal runs you chose. If you are looking at the period 1961-2018, did you select all runs that include those years regardless of the lead time, or is the last run you selected the one that was initialized in 2008?

> We always select the maximum number of possible runs for each lead year range. This means that, for example, the last run used for a lead year 1 evaluation is the one initialized in November 2017, whereas for a lead year 10 evaluation the last run would be the one initialized in November 2008. For longer averaging periods, the last lead year is decisive, so the lead year 4-10 evaluation considers all runs up to 2008, whereas the lead year 4-6 evaluation includes runs up to 2012.
We added a sentence to the end of the section on decadal hindcasts to clarify the choice of runs.

**Methods**

**M1** *Lead times, part 1:* The selection of lead times seems somewhat arbitrary. Why did you choose 4-10 and 7 and not 1-7 and 4 or 2-8 and 5 ...? Have you checked whether your results/conclusions would be different with a different choice of lead time?

> We thank the reviewer for raising this issue. We are aware that choosing lead years 4-10 and 7 is quite arbitrary, and it would be equally valid to choose 2-8 and 5, or 1-7 and 4. We also checked other combinations of the same averaging period and found that similar general conclusions can be drawn from these lead times (see Fig. 1.1). We added a statement to the section on lead times that for reasons of brevity we just show one example for short (7) and long (4-10) averaging periods, respectively, but the conclusions hold for other lead time combinations as well. We also added similar reminders to the results section where we saw fit.

[Figure]

**Fig. 1.1:** Gridpoint-wise anomaly correlation coefficient (ACC) for DJF MSLP anomalies between the hindcast ensemble mean and ERA5 for lead years 1-7 **(a)** and lead year 4 **(b)**. Stippling indicates significant correlations (p<=0.05), determined through a 1000-fold bootstrapping with replacement.

**M2** *Lead times, part 2:* In L126ff, you state that you focus on lead years 4-10 and 7. However, this only applies to the MSLP anomalies, since you show all possible lead year ranges for GBSA. Please be more specific in this regard.

> We clarified in the section on lead times that we only show lead years 4-10 and 7 for MSLP, but all combinations for GBSA.

**M3** *Pressure reduction:* Is this a standard procedure for calculating MSLP from modelled surface pressure? Could you add a reference for equation 1? Does it affect the comparability of your results if you use direct MSLP for one half of the ensemble and calculate MSLP for the other half?

> We added a reference for the pressure reduction formula, which is based on the US standard atmosphere and a fixed air density, as described in Alexandersson et al. (1998) and Krueger et al. (2019). We also performed a consistency check (Fig. 1.2) to quantify the MSLP difference between

direct and derived output and found that it is negligible for low elevations. We therefore added a sentence to the respective section to clarify that we checked the model output for consistency.

[Figure]

**Fig. 1.2:** Difference between manually reduced MSLP and model-output MSLP for one exemplary ensemble member, shown as a 10 year mean (2021-2030) of data from the 2020 initialization. Red colors indicate regions where the manual reduction results in higher MSLP than the automated model output.

**M4** *Region of interest:* Please clarify that you are analysing MSLP anomalies for the entire North Atlantic basin (including the German Bight), whereas the GBSA analyses focus only on the German Bight.

> We thank the reviewer for making us aware that this is unclear. We clarified this paragraph and now explicitly state that we investigate MSLP for the whole North Atlantic basin including the German Bight.

**M5** *Selection of grid points (L140-144):* This information refers to the generation of GBSA time series, correct? If so, either integrate it in the respective paragraph (L146ff) or clarify why you need to select three grid points. At the moment, the whole paragraph comes a bit out of nowhere, without a clear link to the preceding/subsequent paragraphs...

> We thank the reviewer for this suggestion. We agree that the structure of this part of the method section needs improvement to make it more comprehensible. Based on this suggestion and comments from other reviewers, we restructured the paragraphs on the derivation of GBSA from observations.

**M6** *Generation of GBSA time series:* Did I understand correctly that the time series cover the whole period 1960-2018, while you only use the period 1961-2010 for the standardization?

> That is correct. We base the choice of 1961-2010 as a reference period on Krieger et al. (2020), who also used 1961-2010 to standardize the timeseries. We decided to adapt this reference period in order to introduce as few inhomogeneities as possible.

**M7** *Prediction skill:* Please add a short explanation of why it is important to consider both deterministic and probabilistic skill scores when assessing the skill of a decadal prediction system.

> We thank the reviewer for this suggestion. We added two sentences on the importance of both types of predictions to the beginning of the "Evaluation of Model Performance" section.

**M8** *ACC:* Although this should be common knowledge, please add the possible range of ACC and an explanation of what the different values mean.

> We added a sentence on the characteristics of the ACC and the possible range in the respective paragraph.

**M9** *ACC versus BS:* Be careful when using f and o in equations 2 and 4. You chose the same letters, but they have different meanings (value for ACC, probability for BS). Consider replacing f and o in equation 4 with capital letters.

> We thank the reviewer for making us aware of this unclear nomenclature. We changed the variables in Equation 4 and the subsequent paragraph to capital letters to avoid further confusion.

**M10** *Choice of BSS:* Out of curiosity – why did you choose the BSS rather than the ranked probability skill score (RPSS)? Since you are interested in three categories (low/normal/high), the RPSS seems the more natural choice to me as it also contains some information about the distance between model and observations.

> We chose the BSS instead of the RPSS as we wanted to investigate whether the model is particularly skillful in predicting one of the three defined categories. By calculating three distinct skill scores for the dichotomous forecasts *high/not high, low/not low, moderate/not moderate*, we want to demonstrate the added value for forecasts of high activity periods, and the inability of the model for forecasts of moderate activity periods. This distinction would not have been possible by calculating the RPSS, which incorporates the skill for every distinct category into one single measure.
We added additional explanation behind our intention to use the BSS instead of the RPSS in the respective paragraphs and expanded on the differences between the two metrics in the discussion section.

**Results**

**R1** *Some thoughts on L234-242:* Could it be that the initialisation has a "negative" impact in the first years (some kind of initialisation shock) – which would explain why the predictive skill is highest for lead time ranges starting in year 3 and 4? This would also fit (to some extent) to previous studies on wind-related variables like Kruschke et al. (2014) or Moemken et al. (2016). However, these studies use uninitialized historical simulations as reference and not persistence…
For temperature, several studies show high predictive skill for later/longer lead times (e.g. Feldmann et al., 2019). This increase seems to originate mainly from the longterm climate trend. However, I have never heard of the importance of climate trend for decadal predictions of wind-based parameters…

> We thank the reviewer for their thoughts on a possible initialization shock. In fact, the predicted geostrophic winds are lowest for lead years 3-5, and highest for lead year 1 (Fig. 1.3). While we use lead year 1 means and standard deviations to derive standardized GBSA from the absolute geostrophic winds for all data, we also tested whether standardizing each lead year with its respective mean and standard deviation has a notable effect on the results. We find that the ACC between model and observation is almost unaffected by the choice of our standardization reference (Fig. 1.4).
Nevertheless, we expanded the discussion of our results with a paragraph on the effects of a possible initialization shock.
Regarding the climate trend, we agree that the prediction skill for longer lead times can be greatly impacted by the presence of a trend. However, as the reviewer already correctly states, there is little agreement on the response of storm activity to future climate change. Additionally, observational records indicate that, so far, there has not been a significant climate signal in storm activity in our study region, which leads us to believe that long-term trends don't play a major role in the prediction skill here.

[Figure]

**Fig. 1.3:** Absolute 95th annual percentiles of predicted geostrophic winds per lead year. Red dots represent individual members and initialization years, black dots show the ensemble mean for each lead year.

[Figure]

**Fig. 1.4:** ACC between observations and ensemble mean predictions of German Bight storm activity (GBSA) for all combinations of start and end lead years. **Left:** GBSA predictions are based on lead years that are individually standardized by their respective means and standard deviations, i.e., absolute geostrophic winds for lead year 5 are standardized by subtracting the mean and dividing by the standard deviation of lead year 5. **Right:** GBSA predictions are always based on a standardization with respect to lead year 1, i.e., absolute geostrophic winds for lead year 5 are standardized by subtracting the mean and dividing by the standard deviation of lead year 1, like in the original manuscript.

**R2** *L304-338:* These paragraphs seem to be more of a general discussion of your results and are not really related to the rest of section 3.2.2. Therefore, it might make sense to introduce a new section (3.3 Discussion) or new chapter (4. Discussion) for this part of the manuscript.

> We created a separate discussion section where we discuss our results.

**R3** *Persistence as reference:* Many studies dealing with decadal prediction systems use uninitialized historical simulations of the same model or simple climatology as reference. Is there any particular reason why you have not tried this as well? Please do not get me wrong – I think it is a strength of your study that you consider persistence and random guessing. It just makes it harder to compare your results with other studies on decadal predictions.

> After revisiting the manuscript and reviews, we also saw the need to discuss the performance of the model against climatology, as climatology proves to be a tougher challenge than random guessing. We originally opted for persistence and random guessing to not overload the manuscript with a large number of different comparisons, but we agree that using climatology as an additional reference simplifies the comparison of our results with those from other studies. We restructured the results section, added climatology as a reference, and removed the sections and plots discussing the coin-flip-based random guessing.

**Figures**

**F1** For readers unfamiliar with Germany (and the German Bight in particular), it might be helpful to include a figure showing the region of interest. In this, you could also mark the grid points given in Table 1.

> We added a map of the German Bight that shows the location of the triangle to the methods section.

**F2** Figure 2: Please add some explanation in the text (L226-230) about the structure of the plot (that it shows all possible lead time combinations etc.).

> We added a short introduction to the structure of the matrix plots before summarizing the key findings of Figure 6 (Figure 2 in the old manuscript).

**F3** Consider simplifying the captions of Figures 5 and 6 (the same applies to B3 and B4) by saying something like "Same as Figure 4, but for ...".

> We simplified the respective figure captions.

**Specific comments**

> We addressed all minor comments noted by the reviewer as suggested.
* * *
**Response to Reviewer #2**

**2.1 Conclusions**

**2.1A** On the effect of autocorrelated time series on increased correlation coefficients for longer averaging periods

> We agree with the reviewer. We added a paragraph to the discussion section with additional thoughts on the effect of averaging window length on the correlation of associated time series. Please also see our reply to comment 2.4.1B.

**2.1B** On the choice of reference forecasts and the effect of estimating correlations from smoothed timeseries

> We agree that there is a need to discuss the effect of the choice of reference in greater detail. We improved the discussion section to include a comment on the lead-time dependence of persistence (comment 2.4.3A). We also replaced the coin-flip-based random guessing with climatology and discussed this choice of reference as well (more details in comment 2.4.3C).

**2.1C** Revisiting the conclusion that is based on the choice of the Brier Skill Score instead of the RPSS

> The reviewer is correct in assuming that we refer to the RPS/RPSS when mentioning "highly aggregated probabilistic skill scores". We added a paragraph to the newly formed discussion section to highlight our intent and elaborate more on the differences between the general concepts of the RPS and the BS, which we also explain in our reply to comment 2.4.2A.

**2.1D** On a possible initialization shock or model drift

> We thank the reviewer for suggesting the possibility of an initialization shock or model drift. In fact, the predicted geostrophic winds are lowest for lead years 3-5, and highest for lead year 1 (Fig. 2.1). While we use lead year 1 means and standard deviations to derive standardized GBSA from the absolute geostrophic winds for all data, we also tested whether standardizing each lead year with its respective mean and standard deviation has a notable effect on the results. We find that the ACC between model and observation is almost unaffected by the choice of our

standardization reference (Fig. 2.2). Nevertheless, we expanded the discussion of our results with a paragraph on the effects of a possible initialization shock.

[Figure]

**Fig. 2.1:** Absolute 95th annual percentiles of predicted geostrophic winds per lead year. Red dots represent individual members and initialization years, black dots show the ensemble mean for each lead year.

[Figure]

**Fig. 2.2:** ACC between observations and ensemble mean predictions of German Bight storm activity (GBSA) for all combinations of start and end lead years. **Left:** GBSA predictions are based on lead years that are individually standardized by their respective means and standard deviations, i.e., absolute geostrophic winds for lead year 5 are standardized by subtracting the mean and dividing by the standard deviation of lead year 5. **Right:** GBSA predictions are always based on a standardization with respect to lead year 1, i.e., absolute geostrophic winds for lead year 5 are standardized by subtracting the mean and dividing by the standard deviation of lead year 1, like in the original manuscript.

**2.2 Terminology**

**2.2A** On the usage of the term "skill"

> The reviewer is correct that the correlation coefficient per se is not a measure of forecast skill, but much rather a measure of linear association. We replaced the term skill by simply referring to the ACC instead.

**2.2B** On the usage of the terms "deterministic skill" and "probabilistic skill"

> We agree that the terms "deterministic skill" and "probabilistic skill" are not precise, as "deterministic" and "probabilistic" refer to the forecast types. We changed the wording and now refer to the skill of the (probabilistic) prediction instead.

**2.3 Structure**

**2.3A** On the structure of the section on pressure reduction and the derivation of GBSA

> We agree that the title of this subsection needed to be changed to reflect its scope more accurately. We changed the title of the section to "Geostrophic Wind and German Bight Storm Activity" as suggested. We also restructured the entire section and added details to various aspects of the methodology.

**2.3B** On subdividing the model evaluation section into two parts.

> We agree that dividing this section makes it clearer to the reader that we are introducing two different concepts here. We split up the section into two parts, one on the ACC and one on the BSS.

**2.3C** On establishing a separate discussion section.

> We agree that the text from 304 onwards discussed the results rather than describing them. We therefore created a separate discussion section.

**2.4 Statistical concepts**

**2.4.1 Anomaly correlation**

**2.4.1A** On the effect of autocorrelation on significance

> We agree on this point. Calculating confidence intervals and significance levels via the Fisher-z transformation requires independent samples, an assumption that is not satisfied in our case due to autocorrelation. We recalculated the significance with a block-bootstrapping approach, as especially the smoothed (multi-year average) time series are heavily autocorrelated. We also revised the section on the calculation of anomaly correlations accordingly.

**2.4.1B** On the association between forecast and observations and its effect on correlation

> We thank the reviewer for the explanation and sample code on the effect of autocorrelation on the correlation coefficient of smoothed time series. While it doesn't explain the entirety of the ACC increase for longer averaging periods, it might account for a part of it. We added our thoughts on this effect to our discussion section.

**2.4.2 Nature of the probabilistic forecast and Brier score**

**2.4.2A** On the choice of the Brier score instead of the RPS/RPSS

> We thank the reviewer for bringing up the issue of evaluating a 3-category forecast with the Brier (skill) score. We completely agree that the RPS/RPSS is the correct evaluation metric for a 3-category forecast. However, we are not aiming at correctly predicting which category out of the three will occur, but much rather whether the model shows skill for a 2-category forecast with different event thresholds. To use the reviewer's analogy, we are interested how often (out of the three options) the model succeeds in juggling with two balls, not whether the model succeeds in juggling with three balls. While the RPS/RPSS acts as a metric for how well the model performs for a 3-category forecast, it is unable to show whether, for example, a *high vs. no high activity* prediction is more skillful than a *low vs. no low activity* prediction. In our manuscript, the additional skill of the model for high storm activity predictions compared to persistence and climatology would not have been detectable with a single three-category forecast and the RPSS. We totally agree that we needed to clarify this intent better in order to avoid the impression that we aim at generating a 3-category forecast and appreciate your insightful thoughts on this matter. We dedicated a part of the discussion and method sections to our intent and revised paragraphs where our choice of evaluation metric had not been stated clearly before.

**2.4.2B** On the explicit clarification of the "standardization"

> We agree with the reviewer that we needed to clarify our standardization process more explicitly. We rephrased the part of the methods section that introduces the standardization to make this matter clearer.

**2.4.2C** On obtaining the probabilistic forecasts and re-standardization

> We apologize for being vague here. The probabilities are obtained by counting the number of members above/below a category threshold and dividing this number by the total number of members in the ensemble (64). The time series of moving averages for longer periods are standardized again, so that we always compare predicted and observed time series, which by definition have a mean of 0 and a standard deviation of 1. We agree that we needed to describe this more thoroughly, as it indeed makes a difference. We revised the method section and created a separate subsection on re-standardization to reflect this procedure.

**2.4.2D** On the choice of lead years 4-10

> We agree that the choice of lead years 4-10 (and 7) appears quite arbitrary. We could have also done the analysis for lead years 1-7 and 4 (Fig. 2.3) and drawn equally valid conclusions from those lead times. We added a statement to the section on lead times that for reasons of brevity we just show one example for short (7) and long (4-10) averaging periods, respectively, but the conclusions hold for other lead time combinations as well. We also added similar reminders to the results section where we saw fit.

[Figure]

**Fig. 2.3:** Gridpoint-wise anomaly correlation coefficient (ACC) for DJF MSLP anomalies between the hindcast ensemble mean and ERA5 for lead years 1-7 **(a)** and lead year 4 **(b)**. Stippling indicates significant correlations (p<=0.05), determined through a 1000-fold bootstrapping with replacement.

**2.4.3 Reference forecast**

**2.4.3A** On the lead-time dependence of the skill of persistence forecasts

> We thank the reviewer for their thoughts on the lead-time dependence of persistence. We calculated the Brier Score for persistence (Fig. 2.4). While there are certain lengths of averaging periods, for which the BS is lower, there is no general decline in BS with increasing lead time or averaging period. We believe that including plots of the Brier Skill for all forecasts and categories would clutter the manuscript. However, we added paragraphs to the newly created discussion section where we discuss the performance of persistence and its effect on the BSS in more detail.

[Figure]

**Fig. 2.4:** Brier Score of persistence for all combinations of start and end lead years for high storm activity **(left)** and low storm activity **(right)**. Reddish colors indicate BS lower (= better) than that of a coin-flip forecast, bluish colors indicate BS higher (= worse) than that of a coin-flip forecast.

**2.4.3B** On random forecasts

> We are sorry for causing confusion by our ambiguous use of the term "random forecast". Our intention was to create a reference forecast that always predicts a fixed probability of 50%, not one that randomly selects one of two outcomes every year. After consideration, we decided to drop the original 50%-"random" forecast from the manuscript entirely and focus on climatology as our second reference prediction instead. We agree that a reference that draws from the historical climatological distribution (as e.g. described in Wilks, 2011) reflects a random prediction more accurately. Please also see our reply to comment 2.4.3C below.

**2.4.3C** On climatology as a reference forecast

> After revisiting the manuscript and taking into account the points raised by multiple reviewers, we agree that there was the need to compare the model to a climatology-based prediction. We originally opted against that to avoid an excessive number of different references in the results section, but we see that this is a more challenging test for the model than a fixed 50% probability forecast. We revised the results by adding climatology as our second reference prediction and removed the results and discussions related to the fixed 50% prediction. In this process, we also revised the methodology, discussion, and conclusion sections to fit the newly included climatology prediction.

**2.5 Comments on specific lines in the manuscript**

For reasons of brevity, we will refrain from repeating all comments in this document, and instead just refer to the line numbers in the original manuscript.

**9** We thank the reviewer for making us aware of this issue. Our intention behind this statement was to mention that, for certain lead times, a probabilistic forecast can show significant skill, while a deterministic forecast produces insignificant correlation coefficients, and vice versa. We revised the abstract and in the process clarified the confusing statements.

**13** We added an example for the benefit of decadal predictions a.

**32** We addressed the methical difference between evaluating a 3-category forecast with the RPSS (as in Kruschke et al., 2016) and evaluating three separate 2-category forecasts with the BSS and the implications of doing so in the newly created discussion section. Please also see our reply to comment 2.4.2A.

**62** We thank the reviewer for bringing up this additional detail. The uncertainty of a forecast is a great advantage of probabilistic predictions.
With the quoted statement, we intended to explain that periods of high and low storm activity might be detectable through changes in the shape of the ensemble distribution and its tails. If the whole distribution shifts towards one direction, the shift should be notable in both a probabilistic and a deterministic (ensemble mean) prediction. However, if the shape of the tails in particular changes, a probabilistic prediction might hold an advantage over the deterministic prediction. We amended the paragraph to correctly reflect this explanation.

**74** We thank the reviewer for pointing out the difference of using a skill score for deterministic predictions like the MSE and a measure for linear association like the ACC. We believe that the ACC is widely established as a metric to quantify the ability of a climate model ensemble mean to predict the temporal evolution of a quantity. We therefore decided to keep the ACC as our metric but replaced the term "deterministic skill" with ACC (compare comment 2.2A).

**81** That is correct. We standardize time series by applying a z-transformation, i.e., by subtracting the mean and dividing by the standard deviation. We added a clearer definition of the standardization process to the respective paragraphs.

We also added an exemplary histogram of geostrophic wind speeds from one ensemble member, one model run, and one forecast year to illustrate the distribution of geostrophic winds. We also added a violin plot of the distributions of 95th percentiles of geostrophic wind speed in the model ensemble, separated by lead year. For observational GBSA, which is based on an average of 18 different standardized time series of geostrophic winds from overlapping triangles, we do not see fit for such a graphic in the manuscript. The absolute 95th percentiles vary depending on the size of the individual triangles, preventing any generalization of the absolute wind speeds. In order to attempt a rough comparison between model and observations, we compared the range and mean of the modeled 95th percentiles of geostrophic wind speeds to the average 95th percentiles of the observed geostrophic wind speeds from all 18 triangles in Krieger et al. (2020) and found that the mean values agree.

**146** We apologize for leaving this unclear. We use the MSLP gradient of a plane through three grid points, as it was done in previous studies (e.g., Alexandersson et al., 1998; Krueger et al., 2019).  We rephrased this sentence to avoid misconceptions.

**147** We enhanced the explanation of the concept of using standardized 95th percentiles and added an exemplary histogram of absolute geostrophic wind speeds to the respective section.

**154** We thank the reviewer for this suggestion and agree that this title would fit the section better. We updated the title of the section to "Evaluation of Model Performance".

**155** Please see our reply to comment 2.2A.

**166** We amended that sentence to include the dichotomous nature of the Brier score. Please see also our reply to comment 2.4.2A.

**173** We apologize for not going into enough detail here. We bootstrap by sampling different forecast/initialization years with replacement. We do not apply the bootstrap to sample ensemble members. We revised the wording in this paragraph.

**174** We thank the reviewer for pointing out this imprecise wording. We amended the sentence to be more accurate.

**176** We changed the sentence to be more accurate.

**179** The reviewer is correct; our phrasing was ambiguous. We updated this sentence following the suggestion.

**180** We thank the reviewer for this suggestion. We changed the wording and now elaborate more on the characteristics of the Brier Score.

**182** Please see our reply to comment 2.4.3B.

**185** That is correct. We removed the sigma from the category thresholds.

**188** Yes, both GBSA time series from the model and the observations are standardized. We reworded this sentence to clarify.

**189** We apologize for causing confusion here. GBSA is derived from the MSLP gradient of a plane through three grid points, as the reviewer correctly notes. The gradient of the plane can be interpreted as the average horizontal MSLP gradient of the triangle spanned by the three grid points, but there is no averaging of different gradients involved in the derivation of GBSA. We reworded this sentence to avoid misconceptions.

**192** Please see our reply to comment 2.4.3A.

**194** We apologize for being unclear here. The persistence forecast is not a probabilistic forecast, but rather a deterministic one. We use the average storm activity of the past *n* years as a persistence forecast for our target lead years and assign it a Brier Score of either 0 or 1, depending on whether the persistence forecast is on the same side of the threshold as the observation or not. In other words, the persistence always forecasts a probability of either 0% or 100% for an event to happen.
We rephrased the respective paragraphs to clarify that persistence per se is a deterministic prediction that is used as a reference to evaluate the skill of probabilistic predictions of our model.

**204** Please see our reply to comments 2.2A and 2.2B.

**205/209** The reviewer is right; a significant negative correlation is just useful for predictions when the physical reason behind it is clear, which is not the case here. We removed these statements and replaced them by a note that, while the correlation itself is significant, this is of little use for a skillful prediction.

**210** Yes. We rephrased that sentence to clarify that we refer to the absolute correlations being larger on average.

**211** Yes. We thank the reviewer for making us aware of that. We removed the redundant statement.

**212** We thank the reviewer for this suggestion. We added a paragraph on the effect of averaging periods on ACC to the discussion section.

**262** We added thoughts on the performance of persistence to the newly created discussion section. Please also see our reply to comment 2.4.3A.

**263** Please see our reply to comment 2.4.3C.

**265** We apologize for this misleading terminology. We removed the word "absolute", as it does not fit here.

**266** Please see our reply to comment 2.4.3C.

**267** We agree.

**297** Please see our reply to comment 2.4.3C.

**311** We agree with the reviewer that this part of the discussion needed to be enhanced. Please see our reply to comments 2.4.2A and 32.

**316** We thank the reviewer for this suggestion. Please see our reply to comment 2.4.3A.

**323** Please see our reply to comment 2.4.3C.

**325** We agree that it is certainly also a deficit that comes with using persistence as a reference. We think that an improvement in skill over a reference can be seen in two ways, both as a valuable aspect of the DPS and as a deficit of the reference. We expanded on the performance of reference forecasts and the implications in the discussion section.

**335** We thank the reviewer for this thought. Using the 0.16 and 0.84 quantiles would indeed eliminate the need for normal distributed quantities.

**Minor comments**

> We addressed all minor comments noted by the reviewer as suggested.
* * *
**Response to Reviewer #3**

**Comments**

**C1** One issue is that the paper focusses on some unusual forecast lead times (4-10 years and 7 years) without properly motivating why they use these. It would not seem the most interesting lead times for a user of a storm activity forecast. There is frequent reference to short and long averaging periods and I was not always sure whether that referred specifically to these two periods or had been generalised somehow. But if it is the latter, it is not defined. The language needs to be cleaned up around this.

> We thank the reviewer for making us aware of this ambiguity in the manuscript. We are aware that choosing lead years 4-10 and 7 is quite arbitrary, and it would be equally valid to choose 2-8 and 5, or 1-7 and 4. We also checked other combinations of the same averaging period and found that similar general conclusions can be drawn from these lead times (see Fig. 3.1). We added a statement to the section on lead times that for reasons of brevity we just show one example for short (7) and long (4-10) averaging periods, respectively, but the conclusions hold for other lead time combinations as well. We also added similar reminders to the results section where we saw fit.

**C2** Figures which lead to firm conclusions are squirreled away in an appendix. I suggest all important figures need to be in the main paper. See minor comments below.

> We agree that the selection of figures in the main body of the manuscripts needed improvement. We rearranged the figures and, in the process of replacing random guessing with climatology, moved all figures from the appendix to the main body of the paper.

**C3** Another issue is that there is no deterministic skill for mslp anywhere near the German Bight (Figure 1), so how do you explain that you have skill in predicting storm activity there? This needs to be covered in the discussion.

> We apologize for leaving this vague. One way to explain it could be that German Bight storm activity does not depend on the MSLP itself, but on the annual statistics (95th percentiles) of the horizontal MSLP gradients, for which the model shows some skill. This might be due to the model being unable to predict fluctuations around the mean but being able to predict sufficiently large deviations from the mean. We added a paragraph on this contradiction to the discussion section.

[Figure]

**Fig. 3.1:** Gridpoint-wise anomaly correlation coefficient (ACC) for DJF MSLP anomalies between the hindcast ensemble mean and ERA5 for lead years 1-7 **(a)** and lead year 4 **(b)**. Stippling indicates significant correlations (p<=0.05), determined through a 1000-fold bootstrapping with replacement.

**C4** Negative skill is presented as useful skill. It is true, you could multiply the forecast by -1 and get a good forecast on average. The problem is that the skill is possibly negative due to a poorly modelled teleconnection and if there is an individual year when that teleconnection is not strong, multiplying by -1 could be the wrong thing to do. Better to assume negative skill is not useful even if it is significant.

> The reviewer is right; a significant negative correlation is just useful for predictions when the physical reason behind it is clear, which is not the case here. We changed the wording to point out that, while the correlation itself is significant, this is of little use for a skillful prediction.

**C5** Finally, the text often refers to "tails" of the distribution and "extremes" when in fact the data refers to anomalies exceeding 1 sigma, which is neither in the tail or an extreme. These words need to be removed from the text.

> We agree that the terms "tail" and "extremes" can be misleading when talking about +/- 1-sigma events. We reduced the usage of these terms in the manuscript.

**Minor comments**

For reasons of brevity, we will refrain from repeating all comments in this document, and instead just refer to the line numbers in the original manuscript.

**9** Here, "short lead years" should rather read "short averaging periods". We apologize for this mistake and rephrased the abstract accordingly.

**126** We chose two exemplary lead year periods, one for long averaging periods, and one for short averaging periods. There is probably little interest in forecasts for lead years 4-10 specifically, but our intention was to use two cases to bring our point across. We saw the need to motivate this choice better and therefore enhanced the respective paragraphs. Please also see our response to comment C1.

**149** We calculated the means and standard deviations for each member separately to account for possible biases/shifts between individual members, and to force each member to be centered around a storm activity of 0. It would be equally valid to allow biases between members and use the full ensemble mean and standard deviation for the derivation of GBSA. Please see the figure below (Fig. 3.2), showing the model-observation ACC (deterministic skill) after using the individual means and standard deviations (left), as well as after using the full ensemble mean and standard deviation (right). The differences in ACC are negligible. This is also the case for the BSS of the probabilistic forecast.

[Figure]

**Fig. 3.2:** ACC between observations and ensemble mean predictions of German Bight storm activity (GBSA) for all combinations of start and end lead years. GBSA predictions are based on individually standardizing members with their respective means and standard deviations **(left)**, and on standardizing with the mean/standard deviation of the full ensemble **(right)**.

**162** The reviewer is correct. The Fisher-z method requires independent samples, which we did not account for. We recalculated the significance with a block-bootstrapping approach, as especially the smoothed (multi-year average) time series are heavily autocorrelated.
We also agree that adding time series of GBSA would be helpful. However, we want to note that these time series would also likely be for exemplary lead times only, since including time series for all 55 possible lead time combinations would overload the manuscript. Therefore, we decided to add exemplary time series of predicted and observed GBSA for lead years 4-10 and 7.

**195** We rephrased this sentence.

**205/211** Please see our response to comment C4.

**224** We removed "anyhow" from this sentence.

**241** We thank the reviewer for their thoughts on a possible initialization shock. In fact, the predicted geostrophic winds are lowest for lead years 3-5, and highest for lead year 1 (Fig. 3.3). While we use lead year 1 means and standard deviations to derive standardized GBSA from the absolute geostrophic winds for all data, we also tested whether standardizing each lead year with its respective mean and standard deviation has a notable effect on the results. We find that the ACC between model and observation is almost unaffected by the choice of our standardization reference (compare Fig. 3.4 and Fig. 3.2 left). Nevertheless, we expanded the discussion of our results with a paragraph on the effects of a possible initialization shock.

[Figure]

**Fig. 3.3:** Absolute 95[th] annual percentiles of predicted geostrophic winds per lead year. Red dots represent individual members and initialization years, black dots show the ensemble mean for each lead year.

[Figure]

**Fig. 3.4:** ACC between observations and ensemble mean predictions of German Bight storm activity (GBSA) for all combinations of start and end lead years. GBSA predictions are based on lead years that are individually standardized by their respective means and standard deviations, i.e., absolute geostrophic winds for lead year 5 are standardized by subtracting the mean and dividing by the standard deviation of lead year 5, instead of lead year 1 like in the original manuscript.

**246** The reviewer is correct here. If the entire distribution shifts, the shift would also be present in the ensemble mean. The shape of the distribution and specifically the tails needs to change in order for a probabilistic prediction to gain an advantage over the deterministic prediction. We amended the paragraph to correctly reflect this explanation.

**251** We apologize for being unclear. We changed the respective sentences to now precisely state whether we talk about specific lead years, averaging periods, or draw general conclusions.

**260** We added red dots to mark the German Bight on the skill maps.

**271** We thank the reviewer for bringing up this point. We rephrased the respective sections to be more precise in stating that we only show two specific lead year ranges for reasons of brevity but have looked at other lead year ranges as well to draw more general conclusions. We also added a clearer definition of short and long lead year periods in the methods section.

**302** We thank the reviewer for that suggestion. Please see our response to comment C2.

**319** We reworded that sentence to make it less informal.

**323** Absolutely. After revisiting the manuscript and considering the points raised by multiple reviewers, we agree that there is a need to compare the model to a climatology-based prediction, which would be a prediction that uses the climatological probabilities of a year falling into a certain category. We originally opted against that to avoid an excessive number of different references in the results section, but we see that this is a more challenging test for the model than a fixed 50% probability forecast. We revised the choice of reference, removed random guessing from the manuscript, and replaced it with climatology.

**329-338** We created a separate discussion section to discuss our results, as well as limitations and caveats of the model.

**345** That is correct. We updated this sentence.

**361** We expanded on this paragraph and added our thoughts on the origin of the skill for short averaging periods, also taking the performance of the reference prediction into account.

**Fig. 1** We added a map to the methods section that shows the study region and the location of the German Bight triangle.

**References**

Alexandersson, H. et al. (1998): Long-term variations of the storm climate over NW Europe, The Global Atmosphere and Ocean System, 6.

Krueger, O. et al. (2019): Northeast Atlantic Storm Activity and Its Uncertainty from the Late Nineteenth to the Twenty-First Century, Journal of Climate, 32, 1919–1931, doi:10.1175/JCLI-D-18-0505.1.

Kruschke, T. et al. (2016): Probabilistic evaluation of decadal prediction skill regarding Northern Hemisphere winter storms, Meteorologische Zeitschrift, 25, 721–738, doi:10.1127/metz/2015/0641.

Polkova, I. et al. (2019): Initialization and ensemble generation for decadal climate predictions: A comparison of different methods. Journal of Advances in Modeling Earth Systems 11 (1), 149-172, doi:10.1029/2018MS001439.

Wilks, D.S. (2011): Statistical Methods in the Atmospheric Sciences. 3rd Edition, Academic Press, Oxford.

**Additional Changes**

We revised the abstract to include the findings from our comparison with climatology and removed some misleading statements.

We corrected values in the results section that were slightly affected by changing the way of calculating significance.

We reduced the length of the conclusion section, since we believe some of the points fit better into the new discussion session and do not have to be repeated in the conclusion.

We removed the second part of the appendix that contained the comparison with random guessing. As the comparison with random guessing is not part of the manuscript anymore, we see no need to keep this section. Furthermore, all new figures that show the comparison with climatology are now presented in the main body of the manuscript.

In addition, we corrected several typos and grammatical errors throughout the manuscript.
* * *
We would like to thank the reviewers again for their time and effort and their valuable and insightful comments on this manuscript.

We look forward to hearing from you regarding your decision on the manuscript and are happy to respond to any further questions or comments.

Sincerely,

Daniel Krieger
Corresponding Author

---

## Author Response (AR2)

Dear Joaquim Pinto,

Thank you for the opportunity to submit a revised version of our manuscript #EGUSPHERE-2022-288 titled "Skillful Decadal Prediction of German Bight Storm Activity" to *Natural Hazards and Earth System Sciences*. We, the authors, would like to express our gratitude to the two anonymous referees for providing valuable feedback on our manuscript.

We would like to give a point-by-point response to the reviewers' comments in the next section.

**Response to Reviewer #1**

**1** Please merge parentheses in lines 104-106.

We thank the reviewer for this suggestion. We merged the parentheses containing the abbreviations and the ones containing references so that they are consistent with the format used in Section 2.1.

**2** Figure 4 is neither mentioned nor described / discussed in the text.

We apologize for leaving out a statement referencing Figure 4. Our intent was to show an exemplary time series of predicted and observed storm activity for two different lead year periods, as suggested by another reviewer. We added a sentence pointing to the figure and moved the figure into the appendix, because – after some consideration – we believe that the figure is not crucial to understanding the storyline, but rather acts as supplemental information.

**3** 11 figures are quite a lot. Please consider merging Figs 9-11. This would also allow a direct comparison of the different storm activity events.

We agree with the reviewer. We merged Figures 9-11 into one and updated the references in the text.

**4** Please add a reference for the RPSS in line 450.

We apologize for not giving a reference for the RPSS. We added references to the respective paragraph.

**Response to Reviewer #3**

**1** I am now happy with the paper. My concerns have been addressed.

We are delighted to hear that and would like to thank the reviewer again for the constructive feedback.

Again, we appreciate the constructive and valuable feedback by the anonymous reviewers and their dedication to helping us improve our manuscript.

We look forward to hearing from you regarding your decision on the manuscript and are happy to respond to any further questions or comments.

Sincerely,

Daniel Krieger
Corresponding Author